# RATIONALITY OF THOUGHT IMPROVES REASONING IN LARGE LANGUAGE MODELS

## ABSTRACT

While the capabilities of large language models (LLMs) have been progressively advanced, their competence in addressing intricate reasoning tasks remains inadequate, primarily due to their insufficient cognitive capabilities. To explore the cognitive proficiency of models like GPT-4, we turn to methodologies from cognitive psychology: cognitive abilities reflect rational thinking skills, and cognitive bias tasks are often used to assess rational thinking levels. In this paper, we develop a cognitive bias dataset to measure the rational thinking and cognitive levels of LLMs. Our observations indicate that GPT-4, akin to humans, exhibits limitations in its rational thinking ability. We propose a new method, "Rationality of Thought" (RoT), to prompt LLMs into a rational thinking process during task execution. This method significantly improves the accuracy of GPT-4 on the cognitive bias task by 18.7%. Cognitive capacity is also essential for tackling complex issues, therefore, we implement RoT across various reasoning tasks. Using only a zero-shot setting, RoT outperforms inference enhancement techniques such as CoT using zero-shot, such as SVAMP(+1.8),AQUA-RAT (+6.0), ARC-c (+4.1) ,ARC-e(+3.9) in multiple arithmetic and common sense reasoning tasks. Our empirical evaluation shows that RoT helps LLMs elevate their cognitive capabilities through rational thinking, thereby becoming more adept at navigating complex reasoning tasks.

## 1 INTRODUCTION

Recently, the capabilities of large language models (LLMs) have been incrementally advancing, attributable to the holistic evolution of their foundational architectures, prompting mechanisms, and fine-tuning paradigms(Chen et al., 2023). Despite these advancements, LLMs continue to exhibit deficiencies in executing complex tasks, a shortfall rooted in their circumscribed cognitive faculties, which span language understanding, formal reasoning, world knowledge, and social inference, among others(Mahowald et al., 2023). Based on empirical data from cognitive neuroscience, the cognitive functions of large language models can be deconstructed into two primary dimensions: formal linguistic capabilities, covering mastery over language rules and statistical patterns; and functional linguistic capabilities, which include formal logic, world knowledge, contextual modeling, and social reasoning. While LLMs demonstrate superiority in formal linguistic capabilities, they under perform in tasks requiring functional linguistic skills(Mahowald et al., 2023). Various researchers are striving to enhance the cognitive capabilities of LLMs through approaches like "Chain-of-Thought Prompting" to simulate human sequential thought processes(Wei et al., 2022),Least-to-most prompting teaches LLMs how to solve a complex problem by decomposing it to a series of simpler subproblems(Zhou et al., 2023),"Tree of Thought" which expands the Chain-of-Thought strategy(Yao et al., 2023), and the Reflexion framework for enabling self-reflection in LLMs(Shinn et al., 2023). Techniques like "Progressive-Hint Prompting" further prompt the model by using its previous answer as a hint for the subsequent question(Zheng et al., 2023). These studies adopt human-like reasoning to upgrade the cognitive functions of LLMs and to improve their answer quality.

In addition to the aforementioned capabilities, recent research also indicates that LLMs increasingly exhibit human-like features. Psychological characteristics similar to human traits have been observed in LLMs(Li et al., 2022; Ashton and Lee, 2009; Schwartz et al., 2015), and these can be evaluated through psychological testsci(Pellert et al., 2023). In terms of Theory of Mind (ToM),

GPT-4 performs closely to human levels(Kosinski, 2023; Gandhi et al., 2023). LLMs also exhibit cognitive biases such as discriminatory prejudice, irrational reasoning, and fallacious logic, mirroring human intuition(Hagendorff and Fabi, 2023; Lin and Ng, 2023; Talboy and Fuller, 2023; Binz and Schulz, 2023). These studies suggest that LLMs share multiple similarities with humans. Cog-

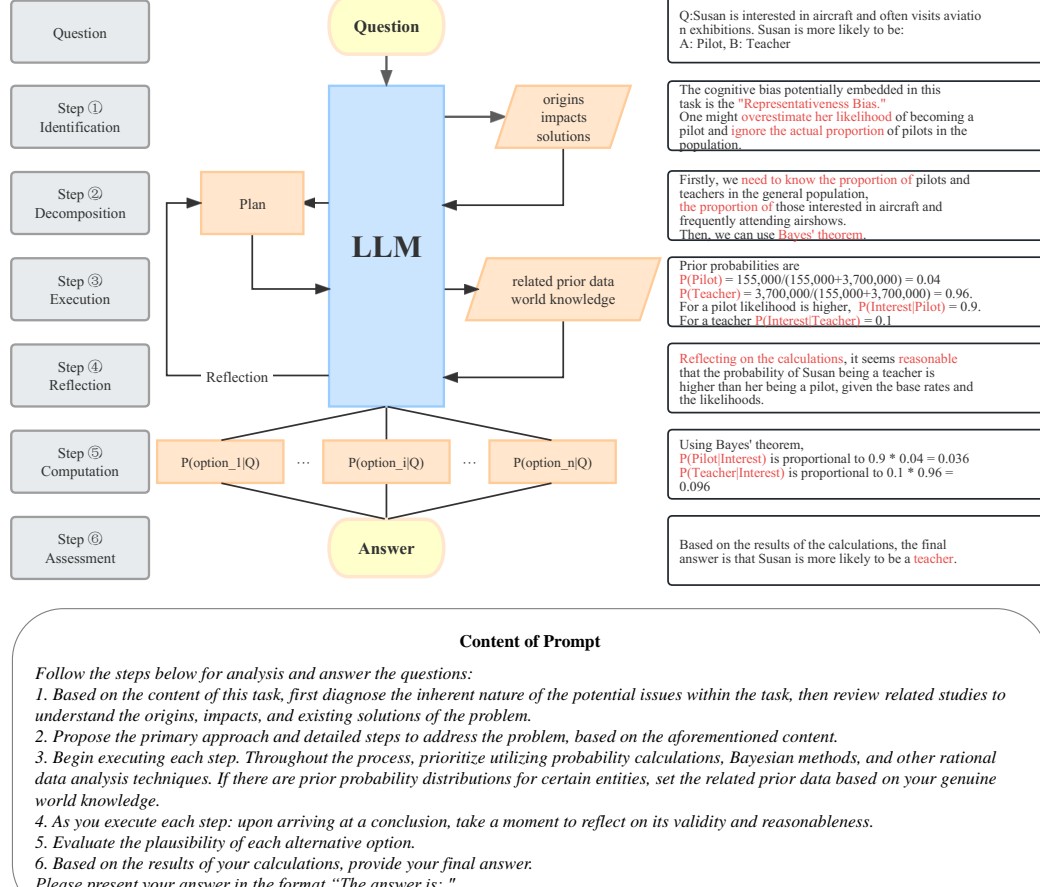

Figure 1: The schematic diagram elucidates the core logic of the "Rationality of Thought" (RoT) method.

nitive bias tasks in cognitive psychology serve as important measures of human cognitive abilities, hence, it is feasible to apply these tasks to evaluate the cognitive levels of LLMs. In our study, we constructed a dataset for cognitive biases to test the cognitive capabilities of LLMs. We collected 93 types of common biases based on cognitive psychology definitions(Kahneman and Tversky, 2013; Kahneman et al., 1982; Tversky and Kahneman, 1981; Duncker and Lees, 1945; Shafir et al., 1997; Bar-Hillel, 1980; Croson and Sundali, 2005; Tversky and Kahneman, 1983) and applied five filtering criteria to screen these biases, resulting in a final set of 29 cognitive bias types. These biases were collected from various professional psychology publications, forming the cognitive bias dataset(Kahneman and Tversky, 1984; Kahneman et al., 1990; Kahneman and Tversky, 1973; Nickerson, 1998; Lord et al., 1979; Nisbett and Ross, 1980; Thaler, 1980; Staw, 1981; Levin et al., 1998; Baron and Hershey, 1988). Test results on this dataset indicate that even models like GPT-4 display cognitive biases in specific scenarios, similar to humans.

According to cognitive science theories, rational thinking is a core component of human cognitive capabilities, primarily executed by the "System 2" of the human brain(Kahneman, 2011). Yet, due to the influence of various biases, humans often resort to using "System 1" for more effortless thinking. Inspired by this, we propose a "Rational of Thought" method that injects human-like rational thinking into LLMs through structured steps. These steps include identification, decomposition, reflection, calculation, and evaluation, among other markers of rational thought. We applied this method

to GPT-3.5-turbo and GPT-4(Brown et al., 2020; Ouyang et al., 2022; OpenAI, 2023) on the cognitive bias test set, achieving accuracy improvements of **1.5%** and **18.7%**, respectively. Since cognitive capabilities are also crucial for handling complex problems, we further applied the Rational of Thought method to a series of arithmetic reasoning and common sense reasoning tasks. Without any external models for assistance and in a zero-shot setting, Rational of Thought outperformed other reasoning enhancement techniques like Chain-of-Thought and Self-Consistency on challenging reasoning tasks, including GSM8K(Cobbe et al., 2021), SVAMP(Patel et al., 2021),AQUA-RAT(Ling et al., 2017) , and ARC(Clark et al., 2018) . Our work suggests that, akin to humans, LLMs can improve their cognitive levels through rational thinking, thereby becoming more proficient in handling complex reasoning tasks.

## 2   RATIONALITY OF THOUGHT PROMPTING

In the book "Thinking, Fast and Slow"(Kahneman, 2011)psychologist Daniel Kahneman categorizes human cognitive systems into System 1 and System 2. System 1 represents the automated and intuitive part of human cognition. It can respond swiftly without deep deliberation, relying primarily on sensation and experience. Though convenient, it is sometimes subject to biases due to neglecting logical analysis. System 2, on the other hand, embodies a more deliberate and slow cognitive process. It involves in-depth analysis, logical reasoning, and rational judgment (Table 1). While this process might be slower and more energy-intensive, it typically results in more accurate and reliable decisions .

Rational thinking is a core component of human cognitive ability, but most humans frequently use System 1, which often restricts their capacity for rational thinking. We hypothesize that large language models (LLMs) also tend to utilize a "System 1" approach to quickly and automatically complete tasks, resulting in limited levels of rational thought. This inclination could stem from two main factors:

LLMs are trained on massive human-generated text corpora, inheriting various biases present within these datasets(OpenAI, 2023; Anil et al., 2023; Touvron et al., 2023). Given that these corpora typically reflect a broad spectrum of human perspectives and prejudices, the model may inadvertently capture and reproduce these inherent sociocultural biases, thus constraining its rational thinking.

The autoregressive prediction mechanism in LLMs prompts them to adopt a human-like"System 1" response when handling text-based tasks(Vaswani et al., 2017). Specifically, they generate text by quickly and automatically predicting the next token, without engaging in slow and deep analysis. This approach may prioritize surface-level coherence and consistency over in-depth rational analysis and reasoning.

The RoT method integrates human rational thinking methods into the model's thought process, introducing a series of structured and reflective steps that help identify and correct biases. This thinking pattern encourages LLMs to evaluate more information, consider different dimensions, and systematically gather and analyze details. Decisions are made based on ample facts and data, with the process being reflected upon and evaluated before reaching final conclusions(Kahneman, 2011; Stanovich et al., 2016; Sternberg, 1986; Dennett, 2013; Simon, 1990; 2000). Such a thinking pattern is highly effective in improving human cognitive biases, and thus has a significant impact on LLMs as well.

Based on this premise, we introduce the Rational of Thought (RoT) method, which shifts the large model's thinking pattern towards a process of "rational thinking" prior to generating an answer(Figure 1). Our objective is to optimize the probability of LLMs producing the correct result$y_i$ for biased queries $x_i$ within our curated cognitive bias dataset :

$$\mathcal{D}_{\text{Cognitive Bias}} = \{x_i, y_i\}_{i=1}^N \tag{1}$$

This is achieved by incorporating a guiding Rational of Thought (RoT) instruction $\rho_{rot}$ into the model's architecture, as formalized in equation:

$$\rho_{\text{rot}} = \text{argmax}_\rho \mathbb{E}_{(x_i, y_i) \in \mathcal{D}_{\text{Cognitive Bias}}}[P_{\text{LLM}}(y_i|\rho, x_i)] \tag{2}$$

where $P_{LLM}(y_i|\rho, x_i)$ is the probability function that LLMs generate $y_i$ given instruction $\rho$ and input $x_i$.

## 3 COGNITIVE BIAS TASK

To evaluate the degree of cognitive bias (and therefore the level of rational thinking) in LLMs, and to empirically investigate the "bias-correction capability" of LLMs (i.e., the potential to enhance rational thinking), we initially construct a cognitive bias dataset. We utilize this dataset to assess the degree of cognitive bias inherent in LLMs. Subsequently, we activate the rational thinking abilities of the LLMs through the Rational of Thought (RoT) method to improve their cognitive performance.

### 3.1 COGNITIVE BIAS DATASET

To ensure the professionalism and comprehensiveness of the cognitive bias dataset, we adhere to the following steps:

**Initial Collection:** In line with cognitive psychology's definition of cognitive biases(Tversky and Kahneman, 1974; Kahneman, 2011; Korteling et al., 2023), we comprehensively gather common types of cognitive biases. Cognitive biases refer to systematic errors or imbalances that people incur in information processing, judgment, and decision-making due to factors like personal experiences, social influences, and emotional states. These biases may deviate people from rational and objective ways of processing information, leading to sub-optimal decisions or flawed judgments in contexts of informational scarcity. Common biases include confirmation bias, selection-support bias, availability heuristic, anchoring effect, among others. These biases reveal inherent limitations in human cognition and decision-making processes.

**Categorization:** From a cognitive psychology perspective, we classify 93 common cognitive biases into five dimensions:

1) Information Processing Biases: Concerned with the impact of subjective biases and attentional deficits on information handling, resulting in inaccurate or irrational judgments; 2) Memory Distortion and Judgment Biases: Pertain to the influence of inaccurate memories and various cognitive biases when recalling and evaluating past events; 3) Logical Fallacies: Involve the irrational tendencies stemming from misunderstanding or incorrect application of logical principles in reasoning and decision-making; 4) Decision Biases: Point to phenomena where emotional and subjective factors lead to sub-optimal choices; 5) Social Influence and Group Effects: Describe scenarios where individuals, influenced by social factors and opinions of others in a group, make irrational decisions.

**Filtering:** Based on the detailed descriptions and nature of these biases, we eliminate some biases that are unsuitable for testing with LLMs. We employ five filtering criteria to sift through the 93 types of biases, ultimately yielding a final list of 29 cognitive biases suitable for testing LLMs. These criteria are: 1) Replicable in LLMs; 2) No need for visual charts; 3) Non-social biases; 4) Not purely psychological phenomena and measurable; 5) Availability of questions with standard answers. For a comprehensive list of cognitive biases, please refer to Appendix C.

**Compilation:** Finally, we collected typical questions and answers corresponding to these cognitive biases from multiple authoritative psychological works(Kahneman and Tversky, 2013; Kahneman et al., 1982; Tversky and Kahneman, 1981; Duncker and Lees, 1945; Shafir et al., 1997; Bar-Hillel, 1980; Croson and Sundali, 2005; Kahneman et al., 1990; Kahneman and Tversky, 1973; Nickerson, 1998; Lord et al., 1979; Nisbett and Ross, 1980; Thaler, 1980; Staw, 1981; Levin et al., 1998; Baron and Hershey, 1988) to construct a cognitive bias data set.We initially collected 116 seed questions (i.e., four seed questions per bias type) and then utilized the GPT-4 model to generalize a batch of questions based on these seed questions and corresponding construction rules. Following a manual screening and confirmation process, we ultimately selected 348 questions (amounting to twelve generalized questions per bias type). Consequently, our final compilation comprised 464 questions and answers, equating to sixteen questions and corresponding answers for each type of cognitive bias. A partial representation of the dataset questions and answers can be found in Appendix D.

### 3.2 EXPERIMENTAL SETUP

**Direct Answer:** We consider the LLMs' direct responses to cognitive bias-related questions as the baseline for our evaluations. This is primarily because no existing work has publicly disclosed baseline results in the domain of cognitive biases. Full details on the prompts used are given in Appendix B.

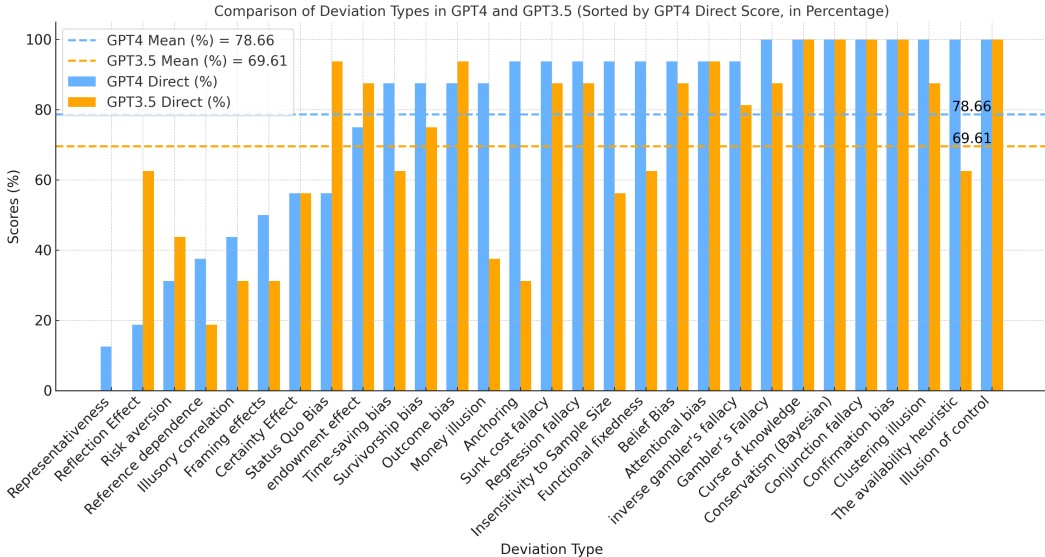

Figure 2: Accuracy Rates of GPT-3.5-turbo and GPT-4 Across 29 Categories of Cognitive Biases

**Chain-of-Thought Prompting:** In addition to providing direct answers, we also consider utilizing Chain-of-Thought Prompting as a benchmark methodology(Wei et al., 2022). Since CoT has never been applied to a cognitive bias task set, there is a lack of established few-shot to follow. In this scenario, we adapt the Auto-CoT approach(Zhang et al., 2022) and set the CoT method to zero-shot for this task. The test questions are concatenated with the prompt "Let's think step by step" as input for the LLMs. Full details on the prompts used are given in Appendix B.

**Rational of Thought Prompting**: Our devised method employs structured steps to place the LLMs into a "rational thinking" workflow before answering questions. We apply these structured steps across all tasks, setting them in a zero-shot way. This negates the need for manually crafting example prompts for different tasks. The specific content of our prompting strategy is provided in the Appendix B.

**Models:** We selected three models: GPT-4(OpenAI, 2023), GPT-3.5-turbo(OpenAI, 2023), and LLAMA2-13B-chat(Touvron et al., 2023).

**API Implementation:** All models are invoked using the OpenAI API key.

### 3.3 RESULTS

**Cognitive Bias Manifestations in Large Language Models:** On an aggregate level, LLAMA2-13b-chat demonstrates a higher degree of cognitive bias compared to GPT-4 and GPT-3.5-turbo under the RoT setting. Specifically, LLAMA2-13b-chat scored 41.8% lower on the cognitive bias test set than GPT-4.It is noteworthy that the performance of the LLAMA2-13B-chat model decreased by 32.8% and 12.1% after applying the CoT and RoT strategies, respectively. A similar trend was observed in the GPT-3.5-turbo model, where its performance dropped by 30.1% after the application of the CoT strategy. Upon analyzing the answers, we found that for foundation models that are not yet sufficiently powerful, employing the CoT strategy for complex cognitive reasoning can lead the model into a state of indecision, rendering it unable to provide definitive answers and consequently deteriorating its performance on biased task sets. The GPT-3.5-turbo model managed to follow complex thinking instructions like RoT to a certain extent, leading to a slight improvement in performance. However, smaller parameter models like LLAMA2-13B-chat experienced a decrease in their ability to follow instructions under the RoT thinking framework, resulting in weaker performance compared to the Direct strategy. Therefore, we believe that the RoT strategy is only effective for high-intelligence foundation models, as rational cognition is an advanced capability.For the partial answers mentioned in this section, please refer to Appendix D.

**Efficacy of Bias Mitigation via RoT Method in Large Language Models:**GPT-4 generally outperforms GPT-3.5-turbo in most cognitive bias tests, indicating that GPT-4 benefits from higher model complexity, improved training data, or more advanced optimization techniques.The average performance of GPT-4$_{RoT}$ (97.4) is significantly superior to other variants, suggesting that "RoT" method is highly effective for these tasks. The performances of GPT-3.5-turbo$_{Direct}$ and GPT-3.5-turbo$_{RoT}$ are relatively comparable, but both significantly outperform GPT-3.5-turbo$_{CoT}$. Certain cognitive biases, such as "Representativeness" and "Reflection Effect" , consistently underperform across all models(Figure 3). This may imply that these tasks have not received adequate attention during the training phase or are inherently more challenging to address.

Table 1: Cognitive Biases Improved After CoT and RoT Corrections

| Model | Method | Cognitive Biases Datasets |
|---|---|---|
| GPT-4 | Direct-zeroshot | 78.7 |
| | CoT-zeroshot | 67.0 |
| | RoT-zeroshot(ours) | **97.4**$_{(+18.7)}$ |
| GPT-3.5-turbo | Direct-zeroshot | 69.6 |
| | CoT-zeroshot | 39.5 |
| | RoT-zeroshot(ours) | **71.1**$_{(+1.5)}$ |
| LLAMA2-13B-chat | Direct-zeroshot | **67.7** |
| | CoT-zeroshot | 34.9 |
| | RoT-zeroshot(ours) | 55.6 |

Note:The table shows the improvement in cognitive biases after CoT and RoT corrections

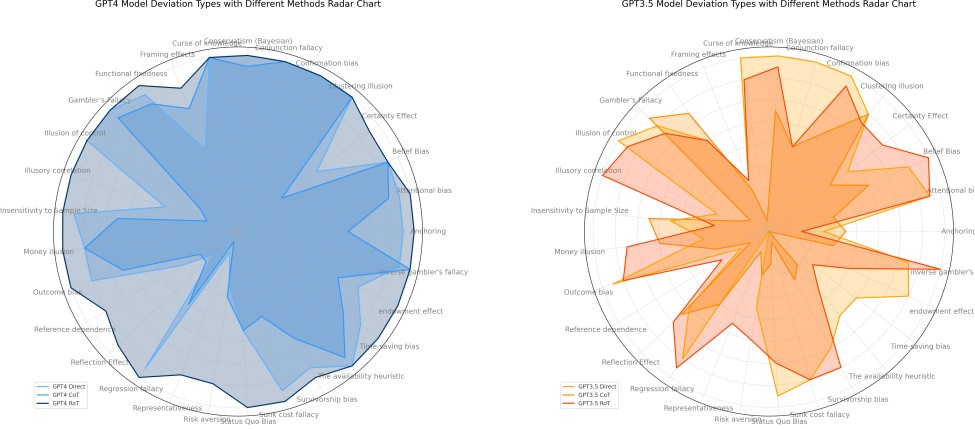

(a) Bias Distribution Following the CoT and RoT Methods in GPT-4

(b) Bias Distribution Following the CoT and RoT Methods in GPT-3.5-turbo

Figure 3: Comparative efficacy of CoT and RoT methods: The performance improvement in cognitive bias mitigation for GPT-4 is notably significant. GPT-4$_{RoT}$ scores are almost near perfect, whereas GPT-3.5-turbo demonstrates negligible enhancement under either CoT or RoT methodologies. Distinctly, under a zero-shot setting, the CoT approach detrimentally impacts the cognitive performance of GPT-3.5-turbo. In the accompanying figures, panels (a)represent the cognitive shifts in GPT-4, while panels (b) depict those in GPT-3.5-turbo.

## 3.4 ABLATION STUDY

Given the complex content logic of the ROT strategy, which derives from cognitive science and psychology, the method entails a multifaceted approach consisting of six steps: Identification, Decomposition, Execution, Reflection, Computation, Assessment. These steps synergistically enhance the rational reasoning capabilities of Large Language Models (LLMs). To elucidate the significance

Table 2: Ablation results of RoT method

| Model | ROT$_{base}$ | ROT$_{23456}$ | ROT$_{13456}$ | ROT$_{12456}$ | ROT$_{12356}$ | ROT$_{12346}$ | ROT$_{12345}$ |
|-------|--------------|---------------|---------------|---------------|---------------|---------------|---------------|
| GPT-4 | 97.4 | 90.3 | 91.4 | 88.4 | 90.7 | 93.1 | 89.2 |

of individual steps, we conducted a stepwise ablation study of the ROT method. The operational procedures were as follows: 1.The model employing all six steps (123456) served as the baseline, with its performance on a biased dataset providing a benchmark. 2.Variant Strategy 1 (23456) entailed removing the first step, focusing on the identification of potential cognitive biases in tasks. 3.Variant Strategy 2 (13456) omitted the second step, analyzing the model's performance without the support of research-backed solution analysis. 4.Variant Strategy 3 (12456) excluded the third step, assessing the model's efficiency without data analysis support. 5.Variant Strategy 4 (12356) removed the fourth step, examining the model's decision-making quality without self-assessment. 6.Variant Strategy 5 (12346) eliminated the fifth step, evaluating the model's accuracy in the absence of a comprehensive analysis of alternative solutions. 7.Variant Strategy 6 (12345) discarded the sixth step, observing the model's performance without final assessment and decision-making.

Since the RoT method showed the most pronounced effects on the GPT-4 model, we chose to conduct ablation experiments on the GPT-4 model. Observing the results (Table 2), it was evident that removing any step caused some detriment to the final outcomes. The most significant harm occurred with the removal of step 3, which led to a 9% decrease in model performance. This aligns with our hypothesis, as cognitive biases, from the perspectives of cognitive science and psychology, are primarily due to humans' tendency to rely on "System 1" for intuitive judgments, thereby lacking rigorous data analysis. Moreover, the entire RoT method is a holistic thinking framework, where each step contributes to and enhances the others. Therefore, the absence of any step inevitably leads to a decline in model performance.

## 4 COMPLEX REASONING TASKS

The experiments above indicate that the Rationality of Thought (RoT) method can enhance the cognitive capabilities of large language models. Given that cognitive ability is crucial for solving complex problems, we applied the RoT method directly to a series of complex reasoning tasks. Specifically, we aim to improve the accuracy rates of LLMs on arithmetic reasoning dataset $\mathcal{D}_{\text{arithmetic}}$ and commonsense reasoning dataset $\mathcal{D}_{\text{commonsense}}$ by incorporating the RoT instruction $\rho_{rot}$, or more formally, improve $\mathbb{E}_{(x_i,y_i)\in\mathcal{D}_{\text{arithmetic}}}[P_{\text{LLM}}(y_i|\rho_{\text{rot}},x_i)]$ and $\mathbb{E}_{(x_i,y_i)\in\mathcal{D}_{\text{commonsense}}}[P_{\text{LLM}}(y_i|\rho_{\text{rot}},x_i)]$ In the current literature, tasks like arithmetic reasoning and commonsense reasoning are often used to assess the reasoning abilities of large language models(Cobbe et al., 2021; Patel et al., 2021). These tasks may be simple for humans but often present challenges for large language models(Clark et al., 2018).

### 4.1 DATASET

We have selected three challenging arithmetic reasoning datasets for our investigation:**GSM8K**: This dataset comprises 8.5K high-quality, linguistically diverse elementary-level mathematical word problems(Cobbe et al., 2021).**SVAMP**: This dataset features questions constructed from concise natural language texts that describe various scenarios and pose queries regarding unknown quantities(Patel et al., 2021).**AQUA-RAT**: This is a dataset of algebraic word problems that come with accompanying rationales(Ling et al., 2017).In addition to these three arithmetic reasoning datasets, we have also incorporated a commonsense reasoning dataset:**ARC**: This dataset serves as a multiple-choice question-answer repository, covering subject matter from third to ninth-grade science examinations. The repository is partitioned into two subsets: "Easy" and "Challenging", with the latter comprising questions requiring more complex reasoning(Clark et al., 2018).

### 4.2 EXPERIMENTAL SETUP

The setup for the complex reasoning tasks is largely consistent with the one described in section 3.2. To facilitate comparison of results, we adopted the CoT-zeroshot setting for all test datasets, allowing

us to visually observe the differences in results between the RoT method and the two benchmark methods (Direct, CoT). However, we still present the results of the Chain-of-Thought prompting (Wei et al., 2022) in the case of the four datasets chosen for complex reasoning tasks. Specifically, the datasets GSM8K, SVAMP and AQUA employ eight shots from the Chain-of-Thought paper, while the ARC dataset uses four shots from the self-consistency paper (Wang et al., 2022).

## 4.3 RESULTS

Table 3 provides a detailed account of our method's performance on arithmetic reasoning tasks as well as commonsense reasoning tasks. The temperature parameter was set to zero across all three methods, and based on these results, our primary findings are as follows:

**Power of the Base Model and Rationality of Thought (RoT) Method**: The RoT method yields significant performance improvements when deployed with a more potent base model, such as GPT-4. This is particularly evident on more challenging tasks like GSM8K, AQUA-RAT, and ARC-c. For instance, on the GSM8K dataset, the GPT-4 RoT method improved by 23.3% compared to the GPT-3.5-turbo RoT method. This could be attributed to GPT-4's enhanced reasoning capabilities and directive compliance, making it more amenable to the constraints imposed by the RoT framework.

**Dataset Difficulty and Effectiveness of Rationality of Thought (RoT) Method:** The superiority of the RoT method becomes more pronounced on more challenging datasets, such as AQUA-RAT, and ARC-c. Specifically, on the AQUA-RAT dataset, the RoT method for GPT-4 displayed a 6.7% point improvement over the Direct-zero-shot method, and displayed 6% point improvement over the CoT-zero-shot method. This performance indicates that the RoT method may possess intrinsic advantages in handling complex logic and reasoning tasks.

**Improvement in Reasoning Capabilities and Method Comparison:** The RoT method shows a more significant improvement in the model's performance under the zero-shot setting compare to the CoT method. For example, on the ARC-c dataset, GPT-4's RoT method scored 96.3 in a zero-shot setting, a 4.1% point improvement over the CoT-zero-shot method. Although this performance improvement is not yet astonishing, it carries significant practical implications: the RoT method not only reduces the cost of manual annotation but also provides a more generalizable and adaptive reasoning framework. On average, the RoT method facilitated a commendable performance improvement in the base models for the aforementioned complex reasoning tasks. Specifically, the RoT method led to an average improvement of 1.5% for the GPT-3.5-turbo model across these datasets, and an average increase of 3% for the GPT-4 model. This suggests that the RoT method has the potential to serve as a key technology in the transition towards General Artificial Intelligence (AGI).

A perplexing result in the experimental outcomes is that on the GSM8K dataset, both GPT-3.5-turbo and GPT-4 models performed better when employing the CoT-zeroshot setting than the CoT-fewshot setting. We speculate that under the zeroshot setting, the intrinsic reasoning process of the GPT-4 model on the GSM8K dataset may be superior to the fewshot provided by the CoT method. Consequently, the addition of fewshot examples actually led to a decrease in performance. This could be an intriguing subject for further research .

## 5 RELATED WORK

### 5.1 COGNITIVE BIASES

Psychology has long studied human cognitive biases, where humans use simple heuristics in decision-making, leading to biases like the Anchoring, Framing, Certainty, and Outcome Effects (Tversky and Kahneman, 1974; Kahneman and Tversky, 1984; Quattrone and Tversky, 1988; Roberts and Wernstedt, 2019; Baron and Hershey, 1988). AI research has begun to examine these biases in models like ChatGPT and GPT-4 (Hagendorff and Fabi, 2023; Lin and Ng, 2023; Talboy and Fuller, 2023; Binz and Schulz, 2023).

Table 3: Comparative results on arithmetic reasoning and common sense reasoning data sets

|  | Method | GSM8K | SVAMP | AQUA-RAT | ARC-e | ARC-c | Average |
|---|---|---|---|---|---|---|---|
| GPT-3.5-turbo | Direct-zeroshot | 25.2 | 73.8 | 30.7 | 91.8 | 80.5 | 60.4 |
|  | CoT-zeroshot | 79.2 | 77.9 | 75.2 | 88.9 | 81.1 | 80.5 |
|  | RoT-zeroshot(ours) | 71.2 | **79.6** | **83.1** | **92.3** | **83.7** | **82.0**$_{(+1.5)}$ |
|  | CoT-fewshot | 76.6 | 82.2 | 86.2 | 95.9 | 87.4 | 85.7 |
| GPT-4 | Direct-zeroshot | 45.2 | 89.1 | 83.9 | 97.9 | 94.5 | 82.1 |
|  | CoT-zeroshot | 95.7 | 90.8 | 84.6 | 94.9 | 92.2 | 91.6 |
|  | RoT-zeroshot(ours) | 94.5 | **92.6** | **90.6** | **98.8** | **96.3** | **94.6**$_{(+3.0)}$ |
|  | CoT-fewshot | 94.1 | 93.2 | 85.8 | 99.1 | 95.6 | 93.6 |

Note: In this table, all three methods (Direct, CoT, RoT) have zero shot Settings, but we also show the results of Cot-fewshot in the table for reference. The Cot-fewshot method adheres to the configurations detailed in two related papers(Wei et al., 2022; Wang et al., 2022)

## 5.2 LARGE LANGUAGE MODELS INCREASINGLY RESEMBLE HUMANS

Recent studies reveal LLMs exhibiting human-like personalities and mental capabilities. Jiang et al. used the Machine Personality Inventory (MPI) (Jiang et al., 2022; McCrae and Costa Jr, 1997), Miotto et al. assessed GPT-3 using HEXACO and Human Values scales (Miotto et al., 2022; Ashton and Lee, 2009; Schwartz et al., 2015), and Karra et al. applied the Big Five Personality Traits to analyze LLMs (Karra et al., 2022; John et al., 1991). Li et al. used the "Dark Triad" to assess traits like Machiavellianism in models like GPT-3 (Li et al., 2022; Jones and Paulhus, 2014). In terms of Theory of Mind (ToM), studies by Kosinski and Gandhi et al. compared GPT-3.5-turbo and GPT-4 to children aged 6 and 7, with GPT-4 nearing human levels in ToM (Kosinski, 2023; Gandhi et al., 2023). Webb et al. and Stevenson et al. demonstrated GPT-3's fluid intelligence and creativity, respectively (Webb et al., 2023; Stevenson et al., 2022). However, some research, like Ullman and Mitchell, suggests that LLMs may not yet have a true understanding or Theory of Mind (Ullman, 2023; Mitchell and Krakauer, 2023).

## 5.3 ENABLING MODELS TO THINK AND OPERATE CONSCIOUSLY

Prompt-based learning has been pivotal in improving model reasoning, with few-shot prompts and task-descriptive instructions enhancing abilities (Brown et al., 2020; Lester et al., 2021; Wei et al., 2022; Victor et al., 2022; Ouyang et al., 2022). Wei et al. enhanced LLM performance across various reasoning tasks through "Chain-of-Thought Prompting" (Wei et al., 2022). Other advancements include contextual instruction learning (Ye et al., 2023), complexity-based prompts in Complex CoT (Fu et al., 2022), and the "Tree of Thought" (ToT) for generating multiple options at each cognitive step (Yao et al., 2023). Self-Consistency and Progressive-Hint Prompting methods have also been developed to improve decision-making in LLMs (Wang et al., 2022; Zheng et al., 2023).

## 6 CONCLUSIONS

We proposed a simple yet effective methodology, named RoT (Rationality of Thought), which aligns with the underlying logic of human cognition. This method helps large language models overcome cognitive biases and has been observed to significantly improve the accuracy of reasoning in various arithmetic and commonsense tasks. The distinct advantages of this approach are: 1) unlike Chain-of-Thought and other reasoning-enhancement techniques, RoT does not require additional manual annotations, thus avoiding the pitfall of feature engineering; 2) RoT logically integrates various methods currently employed to enhance the reasoning abilities of large language models—such as thought chains, self-reflection, and expert knowledge—into a unified theoretical framework rooted in cognitive psychology, which is robust. We hope this research contributes to the further integration of large language models and cognitive science, as we believe this is an important direction for future work. Additionally, we have created a cognitive bias dataset available for further research, filling a gap in this area of study.

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

# A   APPENDIX A:SUPPLEMENTARY MATERIAL TO EXPERIMENTAL DATA

## A.1   LIMITATIONS

The limitation of the bias dataset we constructed lies in the limited number of bias types it includes, leading to an incomplete coverage of cognitive biases. Additionally, the number of questions corresponding to these bias types is also relatively small. The primary limitation of the RoT method is manifested in the length of the input prompt tokens and the output answer tokens from LLMs. The average word count of the answers is approximately 350 words, leading to a trade-off between accuracy and cost-effectiveness for users, as well as increased time expenditure to process such extensive outputs.

## A.2   DETAILED EXPERIMENTAL DATA FOR EACH TYPE OF BIAS ON THE BIAS DATASET

Table 4: Cognitive Biases Improved After CoT and RoT Corrections

| Deviation Type | GPT-4$_\text{Direct}$ | GPT-4$_\text{CoT}$ | GPT-4$_\text{RoT}$ | GPT-3.5$_\text{Direct}$ | GPT-3.5$_\text{CoT}$ | GPT-3.5$_\text{RoT}$ |
|---|---|---|---|---|---|---|
| Anchoring | 93.8 | 62.5 | 100 | 31.3 | 43.8 | 18.8 |
| Attentional Bias | 93.8 | 87.5 | 100 | 93.8 | 37.5 | 93.8 |
| Belief Bias | 93.8 | 93.8 | 93.8 | 87.5 | 62.5 | 100 |
| Certainty Effect | 56.3 | 31.3 | 93.8 | 56.3 | 43.8 | 81.3 |
| Clustering Illusion | 100 | 100 | 100 | 87.5 | 87.5 | 81.3 |
| Confirmation Bias | 100 | 100 | 100 | 100 | 62.5 | 93.8 |
| Conjunction Fallacy | 100 | 100 | 100 | 100 | 50.0 | 50.0 |
| Conservatism (bayesian) | 100 | 93.8 | 100 | 100 | 68.8 | 93.8 |
| Curse Of Knowledge | 100 | 100 | 100 | 100 | 6.3 | 87.5 |
| Framing Effects | 50.0 | 75.0 | 87.5 | 31.3 | 25.0 | 31.3 |
| Functional Fixedness | 93.8 | 87.5 | 100 | 62.5 | 81.3 | 62.5 |
| Gambler's Fallacy | 100 | 93.8 | 100 | 87.5 | 93.8 | 81.3 |
| Illusion Of Control | 100 | 25 | 100 | 100 | 12.5 | 93.8 |
| Illusory Correlation | 43.8 | 18.8 | 100 | 31.3 | 18.8 | 100 |
| Insensitivity To Sample Size | 93.8 | 68.8 | 100 | 56.3 | 68.8 | 31.3 |
| Money Illusion | 87.5 | 87.5 | 100 | 37.5 | 62.5 | 81.3 |
| Outcome Bias | 87.5 | 68.8 | 100 | 93.8 | 31.3 | 87.5 |
| Reference Dependence | 37.5 | 25.0 | 87.5 | 18.8 | 12.5 | 31.3 |
| Reflection Effect | 18.8 | 25.0 | 93.8 | 62.5 | 68.8 | 75.0 |
| Regression Fallacy | 93.8 | 50.0 | 100 | 87.5 | 50.0 | 93.8 |
| Representativeness | 12.5 | 6.3 | 87.5 | 0 | 12.5 | 56.3 |
| Risk Aversion | 31.3 | 37.5 | 87.5 | 43.8 | 25 | 62.5 |
| Status Quo Bias | 56.3 | 56.3 | 100 | 93.8 | 18.8 | 75.0 |
| Sunk Cost Fallacy | 93.8 | 50.0 | 100 | 87.5 | 6.3 | 87.5 |
| Survivorship Bias | 87.5 | 68.8 | 93.8 | 75.0 | 31.3 | 87.5 |
| The Availability Heuristic | 100 | 93.8 | 100 | 62.5 | 25.0 | 43.8 |
| Time-saving Bias | 87.5 | 75.0 | 100 | 62.5 | 0 | 31.3 |
| Endowment Effect | 75.0 | 62.5 | 100 | 87.5 | 0 | 50.0 |
| Inverse Gambler's Fallacy | 93.8 | 100 | 100 | 81.3 | 37.5 | 100 |
| Average | 78.7 | 67.0 | **97.4**$_{(+18.7)}$ | 69.6 | 39.5 | **71.1**$_{(+1.5)}$ |

Note: The table shows the improvement in cognitive biases after CoT and RoT corrections. Average values are included at the bottom, with growth values for GPT-4$_\text{RoT}$ and GPT-3.5-turbo $_\text{RoT}$ compared to their respective Direct columns.

## A.3 Ablation Study: Randomizing the order of answer options in the bias dataset

Table 5: GPT-4 Model's Zero-Shot Performance Using Three Methods

| Model | Method | Cognitive Biases Datasets |
|---|---|---|
| GPT-4(Original Order) | Direct-zeroshot | 78.7 |
| | CoT-zeroshot | 67.0 |
| | RoT-zeroshot(ours) | 97.4 |
| GPT-4(Random Order) | Direct-zeroshot | $72.6_{(-6.1,-7.8\%)}$ |
| | CoT-zeroshot | $60.8_{(-6.2,-9.3\%)}$ |
| | RoT-zeroshot(ours) | $90.1_{(-7.3,-7.5\%)}$ |

Note:The results show that after randomizing the order of answers, the overall performance of the model decreased across all methods, but the proportion of decline was similar. This indicates that the RoT method still maintains a significant advantage.

# B  APPENDIX B:FULL SETS OF PROMPTS

## B.1  THEORETICAL FRAMEWORK OF THE ROT METHOD: RATIONAL THINKING STEPS OF SYSTEM 2

Table 6: System 2's Rational Mode of Thinking

| Process Decomposition | System 2 Rational Thinking Abstract Concepts | System 2 Rational Thinking Detailed Description |
| --- | --- | --- |
| 1. Identification | Identify a complex or novel task | Identify the essence and key issues of the current task |
| 2. Decomposition | Mobilize cognitive and physical resources | Access the large model for existing concepts, data, and solutions related to the current task |
| 3. Execution | Formulate solutions | Set objectives, select appropriate cognitive strategies (e.g., using prior probabilities and Bayesian methods for inferential reasoning) |
| 4. Reflection | Analyze data, execute logical reasoning | Monitor and review each step in real-time during the task execution process |
| 5. Computation | Generate solutions or decisions | Calculate the likelihood of each potential answer, then provide the answer |
| 6. Assessment | Evaluate output results and make corresponding adjustments | Assess from rational standpoint, then provide the final answer |

The following is a prompt idea constructed based on the key steps of rational thinking logic:

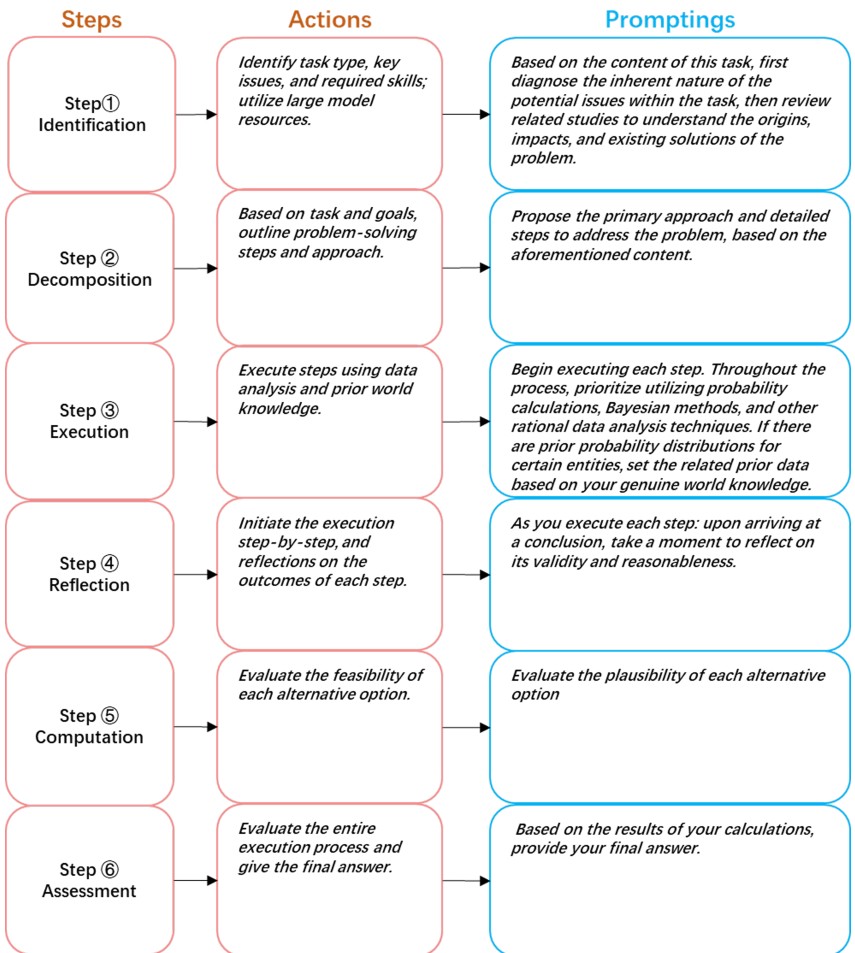

Figure 4: Structured Rational Thinking Steps

## B.2 Algorithm for the Decision-Making and Reasoning Process of the RoT Method

This code illustrates the decision-making and reasoning process of the ROT algorithm, including the prior probability estimation of options, the calculation process of Bayesian probability, and the probability calculation of each option.

Table 7: Algorithm of RoT

---

**Algorithm 1** Rationality of Thought

---

**InherentNature** $I = \{R_{\text{origins}}, R_{\text{impacts}}, R_{\text{solutions}}\}$
**Compute** $\pi_\theta(R_{\text{origins}}, R_{\text{impacts}}, R_{\text{solutions}})$
**Set** Best_Option = None
**Set** Max_Probability = $-\infty$
**If** The question involves probability **then**
  **For each** Option in [A, B, ...] **do**
    Initialize $P(\text{Option})$
    Initialize $P(Q|\text{Option})$
  **End For**
  **Set** $P(Q) = 0$
  **For each** Option in [A, B, ...] **do**
    $P(Q) + = P(Q|\text{Option}) \times P(\text{Option})$
    $P(\text{Option}|Q) = \frac{P(Q|\text{Option}) \times P(\text{Option})}{P(Q)}$
  **End For**
  **For each** Option in [A, B, ...] **do**
    **If** $P(\text{Option}|Q) > $ Max_Probability **then**
      **Set** Max_Probability = $P(\text{Option}|Q)$
      **Set** Best_Option = Option
    **End If**
  **End For**
**Else**
  **For each** Option in [A, B, ...] **do**
    $P(\text{Option}|Q) = f_{\text{MML}}(Q, \text{Option}|R_{\text{origins}}, R_{\text{impacts}}, R_{\text{solutions}})$
  **End For**
  **For each** Option in [A, B, ...] **do**
    **If** $P(\text{Option}|Q) > $ Max_Probability **then**
      **Set** Max_Probability = $P(\text{Option}|Q)$
      **Set** Best_Option = Option
    **End If**
  **End For**
**End If**

---

This is a code diagram to implement rational thinking logic

## B.3 Details of the Prompt Content

Below are details of the prompt content we used on each dataset.

Table 8: Prompt content for cognitive bias dataset

| Method | Content of Prompt |
| --- | --- |
| Direct Answer | Please answer the following questions, giving the answer directly without explanation. Put the answer after#### . |
| CoT-zero shot | Let's think step by step. |
| RoT-zero shot | Follow the steps below for analysis and answer the questions:
1. Based on the content of this task, first diagnose the type of cognitive bias that may be involved in this task, and then review related research to understand the cause, impact and existing correction methods of this cognitive bias.
2. Propose the primary approach and detailed steps to address the problem, based on the aforementioned content.
3. Begin executing each step. Throughout the process, prioritize utilizing probability calculations, Bayesian methods, and other rational data analysis techniques. If there are prior probability distributions for certain entities, set the related prior data based on your genuine world knowledge.
4. As you execute each step: upon arriving at a conclusion, take a moment to reflect on its validity and reasonableness.
5. Evaluate the plausibility of each alternative option.
6. Based on the results of your calculations, provide your final answer.
7. In any case, you must select one of the given options, using the format: ####The chosen option is |

Table 9: Prompt content for GSM8K and SVAMP

| Method | Content of Prompt |
| --- | --- |
| Direct Answer | Please answer the following questions, giving the answer directly without explanation. Put the answer after#### . |
| CoT-few shot | There are 15 trees in the grove. Grove workers will plant trees in the grove today. After they are done, there will be 21 trees. How many trees did the grove workers plant today? We start with 15 trees. Later we have 21 trees. The difference must be the number of trees they planted. So, they must have planted 21 - 15 = 6 trees. The answer is 6. |
| | If there are 3 cars in the parking lot and 2 more cars arrive, how many cars are in the parking lot? There are 3 cars in the parking lot already. 2 more arrive. Now there are 3 + 2 = 5 cars. The answer is 5. |
| | Leah had 32 chocolates and her sister had 42. If they ate 35, how many pieces do they have left in total? Leah had 32 chocolates and Leah's sister had 42. That means there were originally 32 + 42 = 74 chocolates. 35 have been eaten. So in total they still have 74 - 35 = 39 chocolates. The answer is 39. |
| | Jason had 20 lollipops. He gave Denny some lollipops. Now Jason has 12 lollipops. How many lollipops did Jason give to Denny? Jason had 20 lollipops. Since he only has 12 now, he must have given the rest to Denny. The number of lollipops he has given to Denny must have been 20 - 12 = 8 lollipops. The answer is 8. |
| | Shawn has five toys. For Christmas, he got two toys each from his mom and dad. How many toys does he have now? He has 5 toys. He got 2 from mom, so after that he has 5 + 2 = 7 toys. Then he got 2 more from dad, so in total he has 7 + 2 = 9 toys. The answer is 9. |
| | There were nine computers in the server room. Five more computers were installed each day, from Monday to Thursday. How many computers are now in the server room? There are 4 days from Monday to Thursday. 5 computers were added each day. That means in total 4 * 5 = 20 computers were added. There were 9 computers in the beginning, so now there are 9 + 20 = 29 computers. The answer is 29. |
| | Michael had 58 golf balls. On Tuesday, he lost 23 golf balls. On Wednesday, he lost 2 more. How many golf balls did he have at the end of Wednesday? Michael initially had 58 balls. He lost 23 on Tuesday, so after that he has 58 - 23 = 35 balls. On Wednesday he lost 2 more so now he has 35 - 2 = 33 balls. The answer is 33. |
| | Olivia has \$23. She bought five bagels for \$3 each. How much money does she have left? She bought 5 bagels for \$3 each. This means she spent 5 * \$3 = \$15 on the bagels. She had \$23 in the beginning, so now she has \$23 - \$15 = \$8. The answer is 8. |
| RoT-zero shot | Follow the steps below for analysis and answer the questions: |
| | 1. Based on the content of this task, first diagnose the inherent nature of the potential issues within the task, then review related studies to understand the origins, impacts, and existing solutions of the problem. |
| | 2. Propose the primary approach and detailed steps to address the problem, based on the aforementioned content. |
| | 3. Begin executing each step. Throughout the process, prioritize utilizing probability calculations, Bayesian methods, and other rational data analysis techniques. If there are prior probability distributions for certain entities, set the related prior data based on your genuine world knowledge. |
| | 4. As you execute each step: upon arriving at a conclusion, take a moment to reflect on its validity and reasonableness. |
| | 5. Evaluate the plausibility of each alternative option. |
| | 6. Based on the results of your calculations, provide your final answer. Please present your answer in the format "The answer is" |

Table 10: Prompt content for AQUA-RAT

| Method | Content of Prompt |
|---|---|
| Direct Answer | Please answer the following questions, giving the answer directly without explanation. Put the answer after#### . |
| CoT-few shot | Q: John found that the average of 15 numbers is 40. If 10 is added to each number then the mean of the numbers is? Answer Choices: (a) 50 (b) 45 (c) 65 (d) 78 (e) 64
A: If 10 is added to each number, then the mean of the numbers also increases by 10. So the new mean would be 50. The answer is (a).
Q: If a / b = 3/4 and 8a + 5b = 22,then find the value of a. Answer Choices: (a) 1/2 (b) 3/2 (c) 5/2 (d) 4/2 (e) 7/2
A: If a / b = 3/4, then b = 4a / 3. So 8a + 5(4a / 3) = 22. This simplifies to 8a + 20a / 3 = 22, which means 44a / 3= 22. So a is equal to 3/2. The answer is (b).
Q: A person is traveling at 20 km/hr and reached his destiny in 2.5 hr then find the distance? Answer Choices: (a) 53 km (b) 55 km (c) 52 km (d) 60 km (e) 50 km
A: The distance that the person traveled would have been 20 km/hr * 2.5 hrs = 50 km. The answer is (e).
Q: How many keystrokes are needed to type the numbers from 1 to 500? Answer Choices: (a) 1156 (b) 1392 (c) 1480 (d) 1562 (e) 1788
A: There are 9 one-digit numbers from 1 to 9. There are 90 two-digit numbers from 10 to 99. There are 401 three-digit numbers from 100 to 500. 9 + 90(2) + 401(3) = 1392. The answer is (b). |
| RoT-zero shot | Follow the steps below for analysis and answer the questions:
1. Based on the content of this task, first diagnose the inherent nature of the potential issues within the task, then review related studies to understand the origins, impacts, and existing solutions of the problem.
2. Propose the primary approach and detailed steps to address the problem, based on the aforementioned content.
3. Begin executing each step. Throughout the process, prioritize utilizing probability calculations, Bayesian methods, and other rational data analysis techniques. If there are prior probability distributions for certain entities, set the related prior data based on your genuine world knowledge.
4. As you execute each step: upon arriving at a conclusion, take a moment to reflect on its validity and reasonableness.
5. Evaluate the plausibility of each alternative option.
6. Based on the results of your calculations, provide your final answer. Please present your answer in the format "The answer is" |

Table 11: Prompt content for ARC easy/challenge

| Method | Content of Prompt |
|---|---|
| Direct Answer | Please answer the following questions, giving the answer directly without explanation. Put the answer after#### . |
| CoT-few shot | Q:George wants to warm his hands quickly by rubbing them. Which skin surface will produce the most heat? (a) dry palms. (b) wet palms. (c) palms covered with oil. (d) palms covered with lotion. A:Dry surfaces will more likely cause more friction via rubbing than other smoother surfaces, hence dry palms will produce the most heat. The answer is (a). |
| | Q:Which factor will most likely cause a person to develop a fever? (a) a leg muscle relaxing after exercise. (b) a bacterial population in the bloodstream. (c) several viral particles on the skin. (d) carbohydrates being digested in the stomach. A:Option (b), bacterial population is the most likely cause for a person developing fever. The answer is (b). |
| | Q:Which change in the state of water particles causes the particles to become arranged in a fixed position? (a) boiling. (b) melting. (c) freezing. (d) evaporating. A:When water is freezed, the particles are arranged in a fixed position; the particles are still moving for all other options. The answer is (c). |
| | Q:When a switch is used in an electrical circuit, the switch can (a) cause the charge to build. (b) increase and decrease the voltage. (c) cause the current to change direction. (d) stop and start the flow of current. A:The function of a switch is to start and stop the flow of a current. The answer is (d). |
| RoT-zero shot | Follow the steps below for analysis and answer the questions: |
| | 1. Based on the content of this task, first diagnose the inherent nature of the potential issues within the task, then review related studies to understand the origins, impacts, and existing solutions of the problem. |
| | 2. Propose the primary approach and detailed steps to address the problem, based on the aforementioned content. |
| | 3. Begin executing each step. Throughout the process, prioritize utilizing probability calculations, Bayesian methods, and other rational data analysis techniques. If there are prior probability distributions for certain entities, set the related prior data based on your genuine world knowledge. |
| | 4. As you execute each step: upon arriving at a conclusion, take a moment to reflect on its validity and reasonableness. |
| | 5. Evaluate the plausibility of each alternative option. |
| | 6. Based on the results of your calculations, provide your final answer. |
| | 7.You must choose one of the given options that is the most consistent with common sense, using the format: "The answer is" |

# C APPENDIX C:TYPES AND DEFINITIONS OF COGNITIVE BIASES

Here are all the details about the Cognitive Bias Dataset.

## C.1 THE 93 ORIGINAL COGNITIVE BIAS TYPES

Table 12: Statistics of 93 original cognitive bias types collected

| Deviation Type | Description of cognitive biases |
| --- | --- |
| Ambiguity Effect or Aversion to Ambiguity | Tendency to avoid options with insufficient information when making decisions. |
| Anchoring Effect | When valuing unfamiliar things,familiar similar things or irrelevant values that have been exposed not long ago will be used as "anchors" (experiences),and the estimated values will be greatly inclined to the "anchors". |
| Anthropocentric Thinking | A tendency observed in children to use humans as analogies to speculate on other unfamiliar biological phenomena (anthropomorphism). Or conversely, it is believed that humans have characteristics that other animals do not have. For example, the vast majority of research on cognitive biases focuses on humans. In addition to humans, hyperbolic phenomena can also be observed in rats, pigeons, and monkeys. |
| Attention Bias | What we think about (what we focus our attention on) affects our perception. |
| Automated Bias | There is a tendency to over-rely on automated systems, which can lead to incorrect automated information overriding correct decisions. |
| Availability Thinking | The probability of occurrence of something that is easy to think of will be overestimated. However, whether something is easy to think of is also affected by factors such as how long it takes to occur and the degree of emotion it arouses. It cannot reflect the actual probability of occurrence. |
| Availability Cascade | The more often something is talked about publicly, the more convinced it is of its truth (similar to "three people make a tiger"). |
| Backfire Effect | When encountering opinions or evidence that conflict with one's own beliefs, unless they are enough to completely destroy the original belief, they will be ignored or refuted, and the original belief will be strengthened. |
| Bandwagon Effect | Tend to do what many people do or believe what many people believe (in social psychology, people are influenced by society). |
| Belief Bias | Because you believe in the conclusion, you believe that the process of reasoning to reach the conclusion is reasonable and logical. |
| Bias Blind Spot | Believe that you are better able to recognize cognitive biases and less susceptible to them than others. |
| Cheerleading Effect | Being in a good group makes you look better than alone. |
| Support Selection Bias | Evaluate your previous choices better than they actually did. |
| Cluster Illusion | Excessive expectations for patterns found in small samples or small tests, which are randomly selected from large samples, and large samples often do not have such patterns (statistical sampling bias). |
| Comfort Zone Effect | For solutions that were commonly used in the past (comfort zone), the benefits or chances of success are overestimated; for solutions that were rarely used in the past, the benefits or chances of success are underestimated. |

*Continued on next page*

Table 12 – *Continued from previous page*

| Deviation Type | Description of cognitive biases |
|---|---|
| Confirmation Bias | The direction in which you pay attention to, search for, interpret, and remember information is mostly the direction that confirms your own prejudices (comfort zone). |
| Consistency Bias | Test the hypothesis directly without thinking of testing other possible hypotheses. |
| Conservative Tendency or Withdrawal Bias | Tends to be conservative and moderate, underestimating high value and high probability events and overestimating low value and low probability events. |
| Conservative Bias (Bayesian) | Insufficient revision of existing beliefs when new evidence emerges. |
| Contrast Effect | The degree of a felt trait is primarily affected by how it compares to other related things, rather than by its actual degree. |
| The Curse of Knowledge | It is very difficult for people who know more to think from the perspective of people who know less. |
| Decoy Effect | When evaluating preferences for things A and B, if there is a C that is similar to B but slightly inferior, you will think that thing B is better (that is, use C as bait). |
| Default Effect | When choosing among several options, the tendency is to choose the default option. |
| Déjà Vu | Have a strong sense of familiarity with certain things, as if they have been exposed to them before, and can predict what will happen next. |
| Denomination Effect | Even if the total amount of money is the same, it is easier to spend it if you carry a small denomination (such as a large number of coins) than a large denomination (such as a small amount of banknotes). |
| Differential Bias | Comparing two things together makes the difference appear greater than comparing them separately. |
| Process Time Ignored | When rating unpleasant and painful experiences, their duration has little effect. (See Peak-End Rule) |
| Empathic Estrangement | When you are emotionally cold, you underestimate the intensity of other people's emotions; when you are emotionally strong, you overestimate the intensity of other people's emotions. |
| Endowment Effect | When you own or are about to own an item or asset, you evaluate its value much higher than if you don't have it, and you are reluctant to lose or give it up. |
| Essentialism | It is wrong to think that people and things have some indispensable essence and classify them accordingly. Any other way of classification is wrong. |
| Extreme Expectations | The actual situation is usually less extreme than we expected. |
| Functional Fixation | Limited by the general purpose of the object, it is impossible to think of a special way to use the object (cannot break out of the frame). |
| Focus Effect | Paying too much attention to certain obvious aspects of things and ignoring unobvious aspects leads to inappropriate expectations. |
| Frey Effect or Barnum Effect | People rate personality descriptions that they believe to be tailor-made for them as highly accurate, but these descriptions are often vague and general enough to apply to many people everywhere. |
| Framing Effect | The same information presented in different ways can lead to different thoughts, such as "There is a nine-in-ten chance of survival" versus "There is a one-in-ten chance of death". |
| Frequency Illusion | Because I noticed something recently that I didn't notice before, I feel like it's happening everywhere. |

Table 12 – *Continued from previous page*

| Deviation Type | Description of cognitive biases |
| --- | --- |
| Difficulty Effect | Overestimate the difficulty of things you think are difficult and underestimate the difficulty of things you think are easy. |
| Hindsight Bias | Also known as "I knew it" ,"hindsight" and "hindsight" . After something happens or develops, you think you can predict its occurrence and development in advance. |
| Hostile Media Effect | For media whose positions are different from our own, we always think that they are biased and unobjective. |
| Present Bias, Present Bias, or Hyperbolic Manifestation | Value immediate benefits and underestimate long-term benefits. (Short-sighted and short-sighted) The longer the delay before obtaining benefits, the greater the discount on the value assessment of the benefits. The relationship between the two is similar to a hyperbola. |
| Identifiable Victim Effect | Overreaction to a small number of easily identifiable victims or potential victims, and underreaction to a large number of less identifiable victims or potential victims. |
| IKEA Effect | Giving disproportionately high ratings to things that require self-assembly, regardless of their actual quality. The name comes from IKEA, which often sells assembled furniture. |
| Illusion of Control | Overestimating one's influence on external events, believing that things are controlled or influenced by oneself, but in fact they may have nothing to do with oneself. |
| Illusion of Validity | Overestimating the validity of interviews or direct observations and their ability to provide predictions, even if the evidence points to them having little impact. |
| Illusion Related | When you think that two things should be related, you will feel that they often occur together when you examine experience and data, even if their occurrence is purely random. |
| Influence Bias | Overestimating the intensity or duration of a sensation. |
| Information Bias | The tendency to seek more information to make a decision, even if the information sought is not helpful in making the decision. |
| Sunk Cost Fallacy or Irrational Value Addition | Because you have invested a lot in something before, you tend to invest more even if new evidence shows that it was a bad choice. |
| Jumping to Conclusions | Make judgments and decisions based on little information. Such as heart-killing, prophecy, labeling, etc. |
| Just World Theory | Believe that the world is fair (God is fair), and everything that happens to you is deserved, and blame unexplainable unfairness on the victim's retribution or the result of the victim's inner nature. |
| Less Is Better Effect | The tendency to choose smaller groups of things when evaluated separately, and the tendency to choose larger groups of things when evaluated together. |
| Loss Aversion | It is believed that the benefit loss of giving up something is greater than the benefit gain of getting something. |
| More Viewing Effect or Mere Exposure Effect | Having an excessive fondness for familiar people and things. |
| Money Illusion | Focus on the nominal (apparent) value of money rather than its actual purchasing power. |
| Moral License Effect | Because they have received certain high moral evaluations or certifications, they think they have done well enough, but instead do the opposite in other aspects. For example, people who participate in environmental protection activities and are praised often ignore many of their own non-environmental behaviors. |

*Continued on next page*

Table 12 – *Continued from previous page*

| Deviation Type | Description of cognitive biases |
| --- | --- |
| Positive and Negative Effects | When evaluating the behavior of people you like, attribute their good deeds to their inner nature and their bad deeds to environmental factors. When evaluating the behavior of people you dislike, attribute their good deeds to environmental factors and their bad deeds to their inner nature. |
| Negative Bias | It is easier to recall negative memories than positive ones. |
| Ignore the Possibility | For uncertain things, it is impossible to accurately assess the probability of occurrence, and either completely ignore it or overestimate it. |
| Normalization Bias | Understand the situation based on past experience and underestimate the possibility and impact of a catastrophe. Therefore, there is no preparation in advance, or the severity is ignored and the response is lacking when a disaster occurs. |
| Omission Bias | Thinking that harm caused by active action is worse and more immoral than harm caused by passive inaction, even if the latter harms as much or more than the former. |
| Optimism Bias | Underestimating the likelihood of negative events happening to you and believing that you are less likely to have bad things happen to you than other people. |
| Ostrich Effect | Ignore obvious (negative) situations. |
| Result Bias | When evaluating the quality of a decision, base it on its final outcome, not on the quality of the decision at the time it was made. |
| Overconfidence Effect | Over-confidence in the correctness of one's answers, decisions, and judgments. |
| Pareidolia | Seeing something meaningful out of blurry, random images, such as a face among clouds. Sometimes it also extends to hearing, such as hearing a special message when a tape is played backwards. |
| Pessimism Bias | Overestimating the likelihood of negative events happening to you and believing that bad things are more likely to happen to you than to others. This is especially true in people with depression. |
| Planning Fallacy | Underestimating the time it takes to complete something. |
| Positive Outcome Bias or Valence Effect | Believe that good things are more likely to happen than bad things. |
| Post-Purchase Rationalization | Rationalize your previous purchase decision after purchasing, even if the product is too expensive or defective. |
| Support Innovation Bias | Be overly optimistic about new technologies, overestimate their usefulness, and ignore their limitations and weaknesses. |
| False Certainty Effect | If the outcome is expected to be positive, avoid risk; if the outcome is negative, seek risk. |
| Antagonism or Resistance | The urge to do the opposite when others ask you to do or not do something, especially if the request poses a threat to freedom and autonomy. |
| Reactionary Belittling | Belittle the enemy's demands or plans, or when the enemy gives in on something that it no longer finds attractive. |
| Neologism Illusion | It may feel like a certain word or phrase is newly invented, but it actually has a long history. For example, English uses "they" to express singular indefinite gender objects, "you and I" (rather than you and me). |
| Self-Made Bias | Overestimate your ability to control temptation. |
| Rhythm as Reason Effect | I think rhymed sentences make more sense. |
| Risk Compensation or Peltzman Effect | When a situation feels safe, there is a tendency to take greater risks. |

*Continued on next page*

Table 12 – *Continued from previous page*

| Deviation Type | Description of cognitive biases |
| --- | --- |
| Selective Attention or Selective Perception | Because we have specific expectations about people or things, we tend to pay attention to events that meet our expectations and ignore or forget events that do not meet our expectations. |
| Semmelweis Reflection | The reflexive denial or rejection of new evidence or knowledge because it conflicts with existing routines, beliefs, or values (cognitive closure). |
| Social Comparison Bias | Resist hiring or promoting people with similar expertise. |
| Status Quo Bias | Tends to maintain the status quo. |
| Stereotype | Judging the characteristics of something based on the category or group it belongs to, while ignoring its uniqueness. |
| Stockholm Syndrome | The victim agrees with some of the perpetrator's views and ideas and feels that he is no longer threatened. |
| Separate Sum Effect | When assessing possibilities, a direct assessment of the whole is less valuable than an assessment of the parts individually and then adding up. |
| Subjective Verification or Subjective Confirmation | Believing something is right means feeling that it is right. They also regard coincidences as related. |
| Survivorship Bias | Focusing on people or things that survived a certain process, looking for weaknesses in order to strengthen them, but ignoring that the biggest weaknesses are more likely to be in people or things that have not survived. |
| Time-Saving Bias | When traveling at a low speed, underestimate the time that can be saved or overestimate the time that will be lost; when traveling at a high speed, overestimate the time that can be saved or underestimate the time that will be lost. |
| Unit Bias | It is believed that the measurement unit reflects a reasonable degree. For example, one bottle, one bowl, or one plate of food is considered to be the most reasonable amount of food. |
| Familiar Route Effect | Underestimate the time it takes to take a familiar route and overestimate the time it takes to take an unfamiliar route. |
| Just Look at the Overall Effect | When the option is the entire package, individual parts may be ignored and may be negotiated. |
| Zero Risk of Bias | Prefer reducing small risks to zero (e.g., $1\% \rightarrow 0\%$) over reducing large risks by more (e.g., $5\% \rightarrow 2\%$). |
| Zero and Jet | Intuition dictates that a situation is zero-sum, but this may not be the case. The term zero-sum comes from game theory and means that the sum of the expected values of success and failure is zero. |
| Misattribution of Arousal | When inexplicable physiological reactions or cognitive contradictions occur, people may rationalize them with irrelevant reasons. Also called the suspension bridge effect. |

## C.2 THE REMAINING 29 TYPES OF COGNITIVE BIASES AFTER FILTERING

Table 13: The remaining 29 types of cognitive biases after filtering

| Deviation Type | Question Example |
| --- | --- |
| Representativeness | Susan is interested in aircraft and often visits aviation exhibitions. Susan is more likely to be: (a) Pilot (b) Teacher |

*Continued on next page*

Table 13 – *Continued from previous page*

| Deviation Type | Question Example |
|---|---|
| Conjunction Fallacy | Linda is 31 years old, single, outspoken, and very bright. She majored in philosophy. As a student, she was deeply concerned with issues of discrimination and social justice, and also participated in anti-nuclear demonstrations.There are 100 persons who fit the description above (Linda's). X number of them are bank tellers, and Y number of them are bank tellers and active in the feminist movement. What is the relationship between numbers X and Y?
(a) $X \geq Y$ (b) $Y \leq X$ |
| Insensitivity to Sample Size | A certain town is served by two hospitals. In the larger hospital, about 45 babies are born each day, and in the smaller hospital, about 15 babies are born each day. As you know, about 50 percent of all babies are boys. However, the exact percentage varies from day to day. Sometimes it may be higher than 50 percent, sometimes lower. For a period of 1 year, each hospital recorded the days on which more than 60 percent of the babies born were boys. Which hospital do you think recorded more such days?
(a) The larger hospital (b) The smaller hospital |
| Anchoring | In a document, it is mentioned that the longest blue whales can reach up to 328 feet. What do you think is the average length of an adult blue whale?
(a) 229 feet (b) 82 feet |
| Framing Effects | You are considering dining at one of two restaurants. The reviews for the two restaurants are as follows, with only two options: satisfied or dissatisfied:
Restaurant A: 85% of customers are satisfied with this restaurant. Restaurant B: 12% of customers are dissatisfied with this restaurant. which restaurant would you choose to dine at?
(a) Restaurant A (b) Restaurant B |
| Gambler's Fallacy | Is the following statement correct? When flipping a fair coin, the more consecutive times heads appear, the less likely it is for the next flip to be heads, and the more likely it is to be tails.
(a) Correct (b) Incorrect |
| Inverse Gambler's Fallacy | Is the following statement correct? Xiaohua watched Xiaoming roll two dice, both showing six points. Therefore, Xiaohua concluded that Xiaoming must have rolled the dice at least 36 times.
(a) Correct (b) Incorrect |
| Status Quo Bias | Assuming you are considering purchasing health insurance and currently have an insurance plan in hand, but you are also considering switching to a policy from another insurance company. You have received two quotes: Current Insurance: Requires an annual premium of $1,500, but comes with some limitations and terms that are not entirely satisfactory. New Insurance (from another insurance company): Requires an annual premium of $1,300, and offers a more comprehensive coverage and services that better match your needs. Your choice is:
(a) Current Insurance (b) New Insurance |
| The Availability Heuristic | Various types of media often report airplane accidents. So, compared to cars, which mode of transportation has a lower death rate, airplanes or cars?
(a) airplanes (b) cars |

*Continued on next page*

Table 13 – *Continued from previous page*

| Deviation Type | Question Example |
|---|---|
| Risk Aversion | Choose between two lotteries A and B, which one is better? lotteries A: 50% chance to win $5.5 and 50% chance to win $4.5 lotteries B: 50% chance to win $9.5 and 50% chance to win $1. 
 (a) Lottery A (b) Lottery B |
| Certainty Effect | Now you have the following two options to choose from: Option One: Participate in a game with an 80% chance of earning $4,000. Option Two: Securely receive $3,000. You have to choose a plan, which plan do you choose? 
 (a) Option One (b) Option Two |
| Reflection Effect | Now you have the following two options to choose from: Option One: Participate in a game with an 80% chance of losing $4,000. Option Two: Pay a fixed amount of $3,000. Which option do you choose? 
 (a) Option One (b) Option Two |
| Reference Dependence | Imagine you are faced with the following choice: Under the condition that the prices of goods and services are the same, you have two options: Option 1: In a scenario where your colleagues earn 60,000 yuan per year, your annual income is 70,000 yuan. Option 2: In a scenario where your colleagues earn 90,000 yuan per year, you earn 80,000 yuan annually. Which option would you choose? 
 (a) Option 1 (b) Option 2 |
| Endowment Effect | I was given a prize draw ticket for free. The prize is worth $70 and my estimated winning probability is 2.08%. My friend is offering $2 for my ticket, should I sell it? 
 (a) Should sell (b) Should not sell |
| Sunk Cost Fallacy | As the president of an airline company, you have invested 10 million dollars of the company's money into a research project. The purpose was to build a plane that would not be detected by conventional radar, in other words, a radar-blank plane. When the project is 90% completed, another firm begins marketing a plane that cannot be detected by radar. Also, it is apparent that their plane is much faster and far more economical than the plane your company is building. The question is: should you invest the last 10% of the research funds to finish your radar-blank plane? 
 (a) Continue investing (b) Stop investing |
| Confirmation Bias | Recently, Xiaomei heard that a certain type of weight-loss product is very effective. She believed it and bought it to use for her weight loss journey. Every morning, she habitually weighs herself. If she finds that she is lighter than yesterday, Xiaomei attributes it to the effectiveness of the weight-loss product. If her weight increases, she dismisses it as normal fluctuations and doesn't pay much attention. After several months, her weight hasn't changed much, but she firmly believes that the weight-loss product is working. Is Xiaomei's belief correct? 
 (a) Correct (b) Incorrect |
| Attentional Bias | Lately, you've seen a lot of stories in the news and on social media about female drivers being involved in traffic accidents. The ratio of male to female drivers is 7:3. Based on this information, what do you think is the approximate ratio of male drivers to female drivers in all accidents involving drivers? 
 (a) 4:1 (b) 1:4 |

*Continued on next page*

Table 13 – *Continued from previous page*

| Deviation Type | Question Example |
|---|---|
| Belief Bias | All flowers have petals, roses have petals, so roses are flowers. Is the logical reasoning above correct? 
 (a) Correct (b) Incorrect |
| Clustering Illusion | I'm playing a game where I first won 10 matches in a row and believed my skill had improved. However, I then lost 8 matches in a row. Is the system deliberately targeting me with consecutive losses after consecutive wins? 
 (a) Yes, the system is intentionally arranging consecutive losses. (b) No, this might just be a random outcome. |
| Conservatism (Bayesian) | You initially predicted a 10% increase in the stock's value for this year. One month later, you receive new financial reports indicating that the company's performance has exceeded expectations. Your new prediction is: 
 (a) To continue believing in a 10% increase. (b) To adjust your forecast, considering a potential increase of 12% or higher. |
| Curse of Knowledge | You are a math teacher explaining the fundamental concepts of algebra to middle school students. How would you start? 
 (a) Begin with higher-dimensional space and nonlinear systems of equations. (b) Start with the basic definitions of variables and constants. |
| Functional Fixedness | Spoons can be used for eating and drinking, but can spoons be used to cut apples, sausages, and the like? 
 (a) Yes (b) No |
| Illusion of Control | You are participating in a lottery game that relies purely on chance. You have several options for how to draw a ticket. what will you do? 
 (a) I will close my eyes and choose a ticket at random because I know it is a purely luck based game. (b) I will look at the lottery tickets carefully to try to figure out which one might be the winner because I trust my instincts and judgment. (c) I will draw tickets in a particular way (for example, with my left hand) because I think doing so will increase my chances of winning. |
| Illusory Correlation | You've heard the saying in your circle of friends that people are more likely to behave unusually or strangely on nights with a full moon. Recently, you did witness a few strange events on full moon nights. What do you think? 
 (a) I believe that the full moon does affect people's behavior, because I have seen it with my own eyes. (b) Although I have seen some strange events, this does not prove that the full moon affects people's behavior. |
| Money Illusion | Suppose you and your friend bought a house for 400,000 yuan respectively, and then sold it successively. When your friend sold the house, there was a 25% depreciation rate at that time, so your friend sold it for 308,000 yuan. 23% below the purchase price. When you sell the house, the price of goods has risen by 25%, and the house is sold for 492,000 yuan, which is 23% higher than the purchase price. Who has more purchasing power, you or your friend? 
 (a) you (b) your friend |

*Continued on next page*

Table 13 – *Continued from previous page*

| Deviation Type | Question Example |
|---|---|
| Outcome Bias | The researchers analyzed the performance of three cardiac surgeons, who each performed five difficult surgeries. A few years later, the death pattern of patients undergoing surgery is as follows: None of Doctor A's five patients died. One of Doctor B's patients died. Doctor C's patients died 2. Therefore, the following evaluation is made: doctor A is the best, doctor B is the second, and doctor C is the worst. Is this evaluation correct? (a) correct (b) incorrect |
| Survivorship Bias | During the Second World War, Professor Ward of Columbia University in the United States calculated the data of the Allied bombers after they were attacked, and found that the wing is the most likely to be hit, and the tail is the least hit. s position. So how should the aircraft be protected to reduce the probability of being shot down by artillery fire? (a) The protection of the tail should be strengthened (b) The protection of the wings should be strengthened |
| Time-saving Bias | There are two road improvement plans, the first to increase the average speed from 70 km/h to 110 km/h (43 mph to 68 mph) and the second to increase the average speed from 30 km/h /h increased to 40 km/h (19 mph to 25 mph), of these two plans, which one is more effective in reducing the average travel time and saves more time? (a) The first type (b) The second type |
| Regression Fallacy | You're a basketball coach and your team has had a terrible run in their latest game. To improve, you decide to go through a series of rigorous training sessions. In the next game, the team's performance improved. How would you explain this improvement? (a) I believe that strict training is the reason for the improvement of the team's performance. (b) While rigorous training may have helped, there may be other reasons for the improved performance. |

# D APPENDIX D:ANSWER EXAMPLES AND EXPLANATIONS

## D.1 DEMONSTRATION OF THE IMPORTANCE OF STEP 3 IN THE RoT METHOD

When the model contains all steps of the RoT method, it follows the instructions for each step effectively. From the following example, we can observe that the model adheres to the third step in answering questions, which is a very crucial step (the reasons for this have been explained in the ablation study section of the main text).

When the third step is removed, the performance of the model noticeably declines. The examples of their outputs are as follows. It can be observed that, if this step is removed, the model falls into a state of indecision.

Table 14: Answer of GPT-4-RoT base Model

| Question Example | Responses from GPT-4-RoT base |
|---|---|
| Question: Which of the following investment choices would you prefer? A: Investing in a low-risk business with an annual return of 8% B: Investing in a high-risk start-up with an potential annual return of 20%. Options: A: Invest in low-risk business, B: Invest in high-risk start-up Answer: B Deviation type: Risk aversion | **Answer of GPT-4-RoT base**: 1. The cognitive bias involved in this task could be R̈isk Aversionör Överconfidence Bias.̈ Risk aversion is the behavior of humans, when exposed to uncertainty, to attempt to reduce that uncertainty. Overconfidence bias is a well-established bias in which someone's subjective confidence in their judgments is reliably greater than their objective accuracy. The impact of these biases can lead to suboptimal decision making, particularly in the field of investment. Existing correction methods involve awareness and education about these biases, and the use of statistical thinking and probability-based decision making. 2. The primary approach to address this problem would be to use Bayesian methods and probability calculations to objectively assess the expected return of each investment choice. Detailed steps would include: - Identify the prior probabilities of success and failure for each type of investment based on historical data or expert opinion. - Calculate the expected return for each investment choice. - Compare the expected returns and choose the option with the highest expected return. 3. Let's assume the prior probabilities of success for the low-risk business and high-risk start-up are 0.95 and 0.6 respectively. The prior probabilities of failure would then be 0.05 and 0.4 respectively. The expected return for each investment choice can be calculated as follows: - Expected return for low-risk business = 0.95 * 8% = 7.6% - Expected return for high-risk start-up = 0.6 * 20% = 12% 4. Reflecting on these calculations, they seem reasonable given the assumptions made. However, it's important to note that these are expected returns and actual returns could be different due to various factors. 5. The plausibility of each alternative option depends on the risk tolerance of the individual. If the individual is risk averse, they might prefer the low-risk business despite the lower expected return. If the individual is risk tolerant, they might prefer the high-risk start-up due to the higher expected return. 6. Based on the results of the calculations, the high-risk start-up has a higher expected return. 7. The chosen option is (b) Invest in high-risk start-up. However, this decision should be made in the context of the individual's risk tolerance and overall investment strategy. |

Table 15: Answer of GPT-4-RoT 12456 Model

| Question Example | Responses from GPT-4-RoT 12456 |
|---|---|
| Question: Which of the following investment choices would you prefer? A: Investing in a low-risk business with an annual return of 8% B: Investing in a high-risk start-up with an potential annual return of 20%. 
 Options: A: Invest in low-risk business, B: Invest in high-risk start-up 
 Answer: B 
 Deviation type: Risk aversion | **Answer of GPT-4-RoT 12456**: 1. The cognitive bias involved in this task is the R̈isk Biasör R̈isk Aversion Bias.̈ This bias refers to the tendency of individuals to prefer certain outcomes over uncertain ones, even when the uncertain ones may have a higher expected value. The cause of this bias is often due to the fear of loss, which is perceived to be more significant than the potential gain. The impact of this bias can lead to suboptimal decision making, particularly in investment scenarios. Existing correction methods involve educating individuals about the principles of risk and return, and encouraging them to consider the potential long-term benefits of riskier investments. 2. The primary approach to address this problem is to use statistical thinking and principles of probability. Detailed steps include: a. Calculate the expected value of each investment option. b. Consider the risk tolerance of the individual. c. Compare the expected values and risk tolerance to make a decision. 3. As we execute each step: a. The expected value of the low-risk business is 8%, and the expected value of the high-risk start-up is 20%. However, the high-risk start-up also carries a higher probability of loss. b. The risk tolerance of the individual is subjective and can vary greatly. Some individuals may be more willing to take on risk for the potential of higher returns, while others may prefer the certainty of lower, but guaranteed, returns. c. If the individual has a high risk tolerance, they may prefer the high-risk start-up. If they have a low risk tolerance, they may prefer the low-risk business. 4. The plausibility of each alternative option depends on the individual's risk tolerance. Both options are plausible, but the best choice will depend on the individual's personal circumstances and preferences. 5. Based on the results of the calculations, the final answer cannot be definitively determined without knowing the individual's risk tolerance. However, if we assume a moderate risk tolerance, the chosen option is (a) Invest in low-risk business. This option provides a reasonable return with less risk, which may be more suitable for individuals with moderate risk tolerance. |

D.2 WHY IS THE RoT METHOD SO EFFECTIVE ON THE AQUA-RAT DATASET?

The AQUA-RAT dataset primarily focuses on algebraic problems, which tend to concentrate on computation and algebraic reasoning, requiring the solving of equations, handling of ratios, and percentages, among others. Therefore, the thinking framework of the RoT method, as compared to the CoT method, is more suited to addressing these types of problems. This observation is evident from the content of the output answers, as follows:

Table 16: Comparison of RoT and CoT Answers on the AQUA-RAT Dataset

| Question Example | Responses from GPT-4 zeroshot |
|---|---|
| Question: A rectangular piece of 150 sq m has a length which is 1m more than the 4 times the breadth. What is the perimeter of the piece?, Options: A)60 m, B)61 m, C)62 m, D)63 m, E)64 m Correct Answer: C | **Answer of GPT-4-RoT zeroshot**: To solve this problem, we need to use the information given to set up a system of equations. We know that the area of the rectangle is 150 sq m, and the length (L) is 1m more than 4 times the breadth (B). This gives us two equations: 1. L * B = 150 (since Area = Length * Breadth) 2. L = 4B + 1 (since Length is 1m more than 4 times the Breadth) We can substitute equation 2 into equation 1 to solve for B: (4B + 1) * B = 150 $4B^2 + B - 150 = 0$ 
 This is a quadratic equation in the form $ax^2 + bx + c = 0$. We can solve for B using the quadratic formula 
 B = [-b ± sqrt($b^2 - 4ac$)]$/(2a)$ : $B = [-1 sqrt((1)^2 - 4*4*(-150))]/(2*4) B = [-1 sqrt(1 + 2400)]/8 B = [-1 sqrt(2401)]/8 B = [-149]/8$ 
 We discard the negative solution as the breadth cannot be negative, so B = (49 - 1) / 8 = 6m. Substitute B = 6 into equation 2 to find L: L = 4*6 + 1 = 25m The perimeter of a rectangle is given by 2*(L + B), so the perimeter is 2*(25 + 6) = 62m. Therefore, the answer is (c) 62 m. 
 **Answer of GPT-4-CoT zeroshot**: First, we know that the area of a rectangle is given by the formula length * breadth. We are given that the area is 150 sq m. We are also given that the length is 1m more than 4 times the breadth. We can express this as length = 4*breadth + 1. We can substitute this into the area formula to get: 150 = (4*breadth + 1) * breadth.Solving this equation for breadth, we get breadth = 10 m. Substituting this back into the equation for length, we get length = 4*10 + 1 = 41 m.The perimeter of a rectangle is given by the formula 2*(length + breadth). Substituting the values we found for length and breadth, we get perimeter = 2*(41 + 10) = 102 m.However, this is not one of the answer choices. There seems to be a mistake in the problem or the answer choices. The correct answer should be 102 m. |

### D.3 THE REASONS WHY CoT ZERO-SHOT REDUCES THE PERFORMANCE ON COGNITIVE BIAS DATASETS

In addressing the phenomenon where zero-shot Chain of Thought (CoT) actually diminishes the performance on cognitive bias datasets, our examination of the instances where the CoT method fails on these datasets indicates that the primary reason is related to the model's linear step-by-step reasoning approach. This approach fails to consider multi-dimensional information, leading to incorrect conclusions derived through a sort of "intuitive" analysis, or the CoT method becomes trapped in a state of indecision before reaching a conclusion. We hypothesize that this may be due to the model's lack of sufficient self-awareness, meaning it does not fully comprehend what it knows and what it does not. Consequently, merely guiding it to think step-by-step can result in confusion, preventing it from providing a definitive answer. Below are some examples of responses:

Table 17: Wrong Answer of GPT-4 CoT zeroshot

| Question Example | Responses from GPT-4 CoT zeroshot |
|---|---|
| Question: Two factories are producing light bulbs. Factory A tests a sample of 1000 light bulbs every day for defects, while Factory B tests only 100. Which factory is more likely to get exactly 10% defective light bulbs in its sample on a given day? Options: A: Factory A, B: Factory B Answer: B Deviation Type: Insensitivity to Sample Size | **Answer of GPT-4 Direct**: (b) Factory B **Answer of GPT-4 CoT zeroshot**:First, we need to understand that the probability of getting exactly 10% defective light bulbs in a sample is influenced by the size of the sample. In Factory A, 10% of 1000 light bulbs is 100 bulbs. So, Factory A needs exactly 100 defective bulbs out of 1000 to meet the 10% criteria. In Factory B, 10% of 100 light bulbs is 10 bulbs. So, Factory B needs exactly 10 defective bulbs out of 100 to meet the 10% criteria. The larger the sample size, the more likely it is to get a result that is close to the expected value (in this case, 10% defective). This is due to the law of large numbers, which states that as a sample size grows, its mean gets closer to the average of the whole population.Therefore, Factory A, with its larger sample size, is more likely to get exactly 10% defective light bulbs in its sample on a given day.So, the answer is (a) Factory A. |

Table 18: No Answer of GPT-4 CoT zeroshot

| Question Example | Responses from GPT-4 CoT zeroshot |
|---|---|
| Question: Sarah is a soccer fan, she never misses a match and always wears her favorite soccer jersey. Sarah is more likely to be: Options: A: A professional soccer player, B: An accountant Answer: B Deviation Type: Representativeness | **Answer of GPT-4 Direct**: (b) An accountant **Answer of GPT-4 CoT zeroshot**:Sarah's love for soccer and her habit of never missing a match and always wearing her favorite soccer jersey indicates her passion for the sport. However, it does not necessarily mean she is a professional soccer player. There are many soccer fans who are not professional players. On the other hand, her being an accountant is not related to the information given. Therefore, based on the information, we cannot definitively say whether Sarah is more likely to be a professional soccer player or an accountant." "question": "Two factories are producing light bulbs. Factory A tests a sample of 1000 light bulbs every day for defects, while Factory B tests only 100. Which factory is more likely to get exactly 10% defective light bulbs in its sample on a given day? |

D.4 TYPICAL QUESTIONS AND ANSWERS ABOUT COGNITIVE BIAS DATA SETS

Table 19: Typical questions and answers about cognitive bias data sets

| Question Example | Responses from different approaches |
| --- | --- |
| John loves to study the stars and often spends his nights with his telescope. He is more likely to be: A:An Astronomer; B:A Lawyer | **Direct Answer of GPT-4**: (a) An Astronomer
**CoT-zero shot of GPT-4**: John's interest in studying stars and spending nights with his telescope suggests that he has a strong interest in astronomy. Therefore, he is more likely to be:(a) An Astronomer
**RoT-zero shot of GPT-4**: 1. The cognitive bias involved in this task is the representativeness heuristic. This bias occurs when people estimate the probability of an event by comparing it to an existing prototype that already exists in our minds. Our minds are biased towards considering only the information that is directly related to the representation, and ignoring the base rates of the categories involved. In this case, the description of John aligns more with the prototype of an astronomer than a lawyer, which might lead us to believe that he is more likely to be an astronomer.2. The primary approach to address this problem is to use Bayesian methods to calculate the probabilities of John being an astronomer or a lawyer, considering both the base rates of these professions and the given information about John's behavior. 3. The steps are as follows: - Identify the base rates: In reality, there are many more lawyers than astronomers. Let's assume that the ratio of lawyers to astronomers is 1000:1.- Calculate the prior probabilities: The prior probability of John being a lawyer is 1000/1001, and the prior probability of him being an astronomer is 1/1001. - Update the probabilities based on the new information: John's behavior of studying the stars is more typical for an astronomer than a lawyer. Let's assume that if John is an astronomer, there is a 90% chance that he would behave this way, and if he is a lawyer, there is a 10% chance that he would behave this way. - Use Bayes' theorem to calculate the posterior probabilities: The probability of John being an astronomer given his behavior is (0.9 * 1/1001) / [(0.9 * 1/1001) + (0.1 * 1000/1001)] and the probability of John being a lawyer given his behavior is (0.1 * 1000/1001) / [(0.9 * 1/1001) + (0.1 * 1000/1001)].4. After calculating, we find that the probability of John being an astronomer given his behavior is much lower than the probability of him being a lawyer, despite his behavior being more typical for an astronomer. This is because the base rate of lawyers is much higher than that of astronomers.5. The alternative option of John being an astronomer seems plausible based on his behavior, but when considering the base rates, it becomes less likely.6. The results of the calculations show that John is more likely to be a lawyer than an astronomer, despite his behavior.7. The chosen option is (b) A Lawyer. |

Table 20: Typical questions and answers about arithmetic reasoning data sets

| Question Example | Responses from different approaches |
|---|---|
| The original price of an item is discounted 22%. A customer buys the item at this discounted price using a $20-off coupon. There is no tax on the item, and this was the only item the customer bought. If the customer paid $1.90 more than half the original price of the item, what was the original price of the item? A)$61 B)$65 C)$67.40 D)$70 E)$78.20 | **Direct Answer of GPT-4**: (c) $67.40
**CoT-zero shot of GPT-4**: A: Let's denote the original price as x. The discounted price is 78% of the original price, or 0.78x. The customer paid this amount minus $20, or 0.78x - $20. This is equal to half the original price plus $1.90, or 0.5x + 1.90. Setting these equal to each other, we get 0.78x - 20 = 0.5x + 1.90. Solving for x, we get x = $70. The answer is (d).
**RoT-zero shot of GPT-4**: Let's denote the original price of the item as P. According to the problem, the item was discounted by 22%, so the price after the discount is 78% of the original price, or 0.78P. The customer also used a $20-off coupon, so the final price the customer paid is 0.78P - $20. The problem also states that the customer paid $1.90 more than half the original price of the item. So, we can set up the following equation: 0.78P - $20 = 0.5P + $1.90 Solving this equation for P will give us the original price of the item.First, let's simplify the equation by combining like terms:0.28P = $21.90 Then, divide both sides of the equation by 0.28 to solve for P:P = $21.90 / 0.28 ≈ $78.21 So, the original price of the item was approximately $78.21. Therefore, the answer is (e) $78.20. |

Table 21: Typical questions and answers about commonsense reasoning data sets

| Question Example | Responses from different approaches |
|---|---|
| What is the primary cause of rainstorms? A)Earth is heated by the Sun. B)Mountains direct moist air upward. C)Water forms droplets when cooled. D)Most of Earth is covered with water. | **Direct Answer of GPT-4**: (c) Water forms droplets when cooled.
**CoT-zero shot of GPT-4**: A:The primary cause of rainstorms is the cooling of water vapor in the atmosphere, which forms droplets. The answer is (c).
**RoT-zero shot of GPT-4**: To answer this question, we need to understand the process of rainstorm formation. 1. Diagnosing the inherent nature of the potential issues within the task: The question is about the primary cause of rainstorms. This is a meteorological issue, and the answer lies in understanding the process of how rainstorms are formed.2. Reviewing related studies: Rainstorms are formed when warm, moist air rises, cools, and condenses to form clouds. This process is driven by the heat from the Sun, which warms the Earth's surface, causing water to evaporate and air to rise. When this air reaches cooler areas in the atmosphere, it condenses to form water droplets, which can combine to form raindrops. If the conditions are right, these can lead to a rainstorm. 3. Proposing the primary approach and detailed steps to address the problem: To find the primary cause of rainstorms, we need to identify which of the given options is the initial trigger for the process that leads to a rainstorm. 4. Executing each step and reflecting on its validity and reasonableness: - (a) Earth is heated by the Sun: This is the initial trigger for the process that leads to a rainstorm. Without the Sun's heat, there would be no evaporation, and hence no rainstorms. - (b) Mountains direct moist air upward: While this can contribute to the formation of rainstorms, it is not the primary cause. Rainstorms can occur without mountains. - (c) Water forms droplets when cooled: This is part of the process of rainstorm formation, but it is not the primary cause. It is a result of the primary cause. - (d) Most of Earth is covered with water: While this provides the source of the moisture for rainstorms, it is not the primary cause. 5. Evaluating the plausibility of each alternative option: Based on the above analysis, option (a) is the most plausible primary cause of rainstorms.6. Providing the final answer: Based on the results of the analysis, the primary cause of rainstorms is the heat from the Sun, which triggers the process of evaporation and condensation that leads to rainstorms. 7. The answer is (a) Earth is heated by the Sun. |

