# OpenReview forum: "Rationality of Thought Improves Reasoning in Large Language Models"
_ICLR.cc/2024/Conference — Submitted to ICLR 2024_

### Official Review · Reviewer_58ny · 2023-10-27

**Soundness:** 3 good
**Presentation:** 3 good
**Contribution:** 3 good
**Rating:** 6
**Confidence:** 4

**Summary:**

The authors propose Rationality of Thought as a new method to prompt LLMs into a rational thinking process. In addition, they propose a new benchmark inspired by problems that have been used to study cognitive biases in humans. Their methods improve the accuracy of LLMs on their new benchmark and on existing reasoning benchmarks (but to a lesser degree).

**Strengths:**

The proposed method is interesting, clean, and well-motivated.

The paper is well-written and easy to follow.

The authors propose a new and interesting benchmark.

The obtained results are promising.

**Weaknesses:**

The authors write that each type of bias has 16 questions in their data set. Were all of these problems taken from prior literature or does this also include novel problems created by the authors? My main concern about the current state of the paper is regarding this issue. Many of the original cognitive bias problems (e.g., the Linda problem) are presumably heavily featured in the training data. To ensure that we are probing the reasoning capabilities of LLMs, it would be important to include novel (i.e., rephrased) versions of the original problems. While the current set does not include any such problems, I am unlikely to increase my score further. Maybe this is already done but the information is just not provided. In this case, the authors should describe how the rephrasing was done.

Important references to prior work are missing. For example, Binz & Schulz (2023) already studied some of the biases included in the present benchmark. In addition, the present paper takes quite a one-sided view in claiming that LLMs are becoming increasingly similar to human reasoning. There are many examples where this is not the case, see for instance Ullman (2023) or Mitchell & Krakauer (2023).

Binz, M., & Schulz, E. (2023). Using cognitive psychology to understand GPT-3. Proceedings of the National Academy of Sciences, 120(6), e2218523120.

Ullman, T. (2023). Large language models fail on trivial alterations to theory-of-mind tasks. arXiv preprint arXiv:2302.08399.

Mitchell, M., & Krakauer, D. C. (2023). The debate over understanding in AI’s large language models. Proceedings of the National Academy of Sciences, 120(13), e2215907120.

The visual presentation could be improved in places. In particular, while Figure 1 gives a good intuitive understanding of the approach, the reader has to go to the SI to see the exact RoT prompt. This is unfortunate as this is the heart of the paper. Furthermore, Table 1 does not add any value in my opinion. A good solution would be to therefore replace Table 1 with Table 5 from the SI or similar.

Minor:

Formatting is a bit weird in places, especially spaces are often used wrongly/inconsistently.

Equation on page 3:
* is not numbered.
* p_rot does not appear.
* \in \mathcal{D} should be in the subscript (also for the equations later in the paper).
* the equation implies that there is some optimization happening but that is not the case to my understanding (maybe I am missing something here).

Figure 3: legend and text are way too small to be readable. To allow for larger font sizes, subplots a and c could be removed given that they display redundant information with b and d.

The conclusion is not a conclusion as it is followed by a related work section.

Page 9: the authors say “Research by Alaina ...” but then cite another author (maybe first and surnames are exchanged here).

**Questions:**

The authors use zero-shot CoT for the experiments in section 3 and few-shot CoT for the experiments in section 4. Why was this choice made? It would have been easy to also use zero-shot CoT for the experiments in section 4 (especially if the claim is that RoT is nice because it does not require example solutions). Related to that, the fact that zero-shot CoT actually decreases performance on the cognitive bias dataset is surprising. Any idea why this is the case?

Given that the improvements were not so great in GPT3.5, would you say that RoT prompting is an emergent ability?

Finally, I was wondering whether all the steps in RoT prompting are needed or whether some of them could be removed without a loss of performance. I am not asking the authors to run additional ablations but I think this is still interesting to think about.

---

> ### Author Response · Authors · 2023-11-22
>
> Dear Reviewers,
>
> First and foremost, we would like to express our sincere gratitude for the time and effort you have dedicated to reviewing our manuscript.Your reviews have been extremely helpful in improving our research work. This process has not only strengthened our current manuscript but also enriched our understanding of the subject matter.
> In the past week, in reference to your reviews, we have supplemented the relevant experiments and clarified some of the previously unclear descriptions in our manuscript. Next, we will respond to your comments one by one.
>
> Please kindly note, due to the extensive nature of my response and the limitation of 5000 characters per message, I will be submitting my response in multiple parts. I kindly ask for your patience and suggest that you review them collectively once all parts have been sent. Thank you very much for your understanding.
>
>
> | Comment | Response |
> | --- | --- |
> | The authors write that each type of bias has 16 questions in their data set. Were all of these problems taken from prior literature or does this also include novel problems created by the authors? My main concern about the current state of the paper is regarding this issue. Many of the original cognitive bias problems (e.g., the Linda problem) are presumably heavily featured in the training data. To ensure that we are probing the reasoning capabilities of LLMs, it would be important to include novel (i.e., rephrased) versions of the original problems. While the current set does not include any such problems, I am unlikely to increase my score further. Maybe this is already done but the information is just not provided. In this case, the authors should describe how the rephrasing was done. | In this study, during the construction of the bias dataset, we initially collected 116 seed questions, corresponding to four seed questions for each type of bias. Subsequently, utilizing these seed questions and associated construction rules, we employed the GPT-4 model to generate a broader set of questions. After manual review and selection, 348 questions were finalized, equating to 12 generalized questions per bias type. This implies that at least 75% of the questions in this dataset were not present in the training data of the models we used (GPT-4 and GPT-3.5), thus ensuring the quality of this bias dataset for validating the inferential capabilities of Large Language Models (LLMs). Detailed information regarding the dataset construction has been added to the latest version of the paper, specifically in Section 3.1, titled "Cognitive Bias Task". |
> | Important references to prior work are missing. For example, Binz & Schulz (2023) already studied some of the biases included in the present benchmark. | Yes, thank you very much for your reminder. We have carefully read the paper and have already incorporated its core content and references into Section 5.1 'Cognitive Biases' of the latest version of our manuscript. |

---

> > ### Author Response · Authors · 2023-11-22
> >
> > | Comment | Response |
> > | --- | --- |
> > | In addition, the present paper takes quite a one-sided view in claiming that LLMs are becoming increasingly similar to human reasoning. There are many examples where this is not the case, see for instance Ullman (2023) or Mitchell & Krakauer (2023). | Thank you for your reminder. After careful consideration, we agree that these works are indeed meaningful for the future research of Large Language Models (LLMs). Therefore, we have integrated the core content of these studies into the appropriate sections of our paper. Specifically, the core content and references of this work have been added to section 5.2, "Large Language Models Increasingly Resemble Humans," in the latest version of our manuscript. However, we would like to clarify our research approach: There is indeed a portion of research that contends LLMs do not possess genuine human cognition. However, it is important to note that: 1. This body of work primarily focuses on tests conducted with GPT-3 and GPT-3.5 models, which do not perform well in experimental tasks. This aligns with our experimental conclusions. We tested GPT-3.5 and LLAMA2-13B (an open-source model released by Meta) using a bias dataset and found that these models performed only moderately well in cognitive bias tasks, with only GPT-4 showing remarkable performance; 2. We believe that the increasing similarity between models and humans is not solely based on cognitive performance, but also on the fact that these models are trained on massive amounts of human-generated textual data. This training imbues the models with inherent human language characteristics (such as biases), and since human language is one of the manifestations of human behavior, we posit that current LLMs are naturally inclined to mirror general human behaviors and characteristics due to their training methodologies. We maintain an open stance regarding the conclusion that LLMs now truly possess human intelligence. This study, by employing a bias dataset to test various models, aims to experiment with and explore some of the most advanced LLMs currently available. We believe this approach holds significant merit. |
> > | The visual presentation could be improved in places. In particular, while Figure 1 gives a good intuitive understanding of the approach, the reader has to go to the SI to see the exact RoT prompt. This is unfortunate as this is the heart of the paper. Furthermore, Table 1 does not add any value in my opinion. A good solution would be to therefore replace Table 1 with Table 5 from the SI or similar. | Thank you for the excellent suggestion. We have moved Table 1 to Appendix B and illustrated the specific prompt content in Figure 1. |
> > | Formatting is a bit weird in places, especially spaces are often used wrongly/inconsistently. | We appreciate your reminder and have rechecked the formatting issues, hoping that there are no errors this time. |
> > | Equation on page 3:is not numbered. p_rot does not appear.\in \mathcal{D} should be in the subscript (also for the equations later in the paper).the equation implies that there is some optimization happening but that is not the case to my understanding (maybe I am missing something here). | We have numbered the main formulas, but some minor ones are directly incorporated into the main body of the paper in Chinese to save space. The term \(\in \mathcal{D}\) has already appeared throughout the text as a subscript in equations. Regarding the confusion about optimization that you mentioned, our explanation is as follows: this study involves adjusting prompts on a biased dataset \(D\) to maximize the expected score. We consider this to be a form of optimization process. |
> > | Figure 3: legend and text are way too small to be readable. To allow for larger font sizes, subplots a and c could be removed given that they display redundant information with b and d. | Your feedback is excellent. We have removed subfigures a and c to enhance the clarity of the image. Thank you very much for your meticulous attention. Please refer to page 6 of the updated version of the paper for these changes. |
> > | The conclusion is not a conclusion as it is followed by a related work section. | We have restructured the paper by moving the conclusion to the final section. Additionally, in order to streamline the paper and save space, we have condensed the content related to prior work in this version (while retaining all citations to relevant work). We believe you will also observe this modification in the new version of the paper. |
> > | Page 9: the authors say “Research by Alaina ...” but then cite another author (maybe first and surnames are exchanged here). | Yes, your guess is correct, only the first and last names were exchanged here, but in order to avoid confusion for the readers, we have modified the names in the main text. |

---

> > > ### Author Response · Authors · 2023-11-22
> > >
> > > | Comment | Response |
> > > | --- | --- |
> > > | The authors use zero-shot CoT for the experiments in section 3 and few-shot CoT for the experiments in section 4. Why was this choice made? It would have been easy to also use zero-shot CoT for the experiments in section 4 (especially if the claim is that RoT is nice because it does not require example solutions). Related to that, the fact that zero-shot CoT actually decreases performance on the cognitive bias dataset is surprising. Any idea why this is the case? | In the previous experimental phase, we utilized the CoT few-shot approach to comprehensively replicate the performance of COT on the same dataset. In the most recent experiments, we have introduced a COT zero-shot strategy. The results reveal that, compared to CoT zero-shot performance, RoT zeroshot exhibits significant improvements for the GPT-4 model on the following datasets: SVAMP (+1.8), AQUA-RAT (+6), ARC-e (+3.9), and ARC-c (+4.1). This enhancement in performance is quite substantial, and based on these findings, we posit that the Rational of Thought method can enhance arithmetic reasoning and common sense reasoning tasks. For detailed data, please refer to Table 3 on page 9 of the latest version of the paper. Regarding the phenomenon of COT zero-shot actually diminishing performance on cognitive bias datasets, our examination of the failed questions in the COT method on cognitive bias datasets has revealed several reasons. Primarily, when the model engages in a step-by-step thought process, it tends to overlook additional dimensions of information. It arrives at incorrect answers through an "intuitive" analysis, or the COT method may become trapped in a "difficult decision" state before providing a conclusive response. We speculate that this could be attributed to the model's lack of strong self-awareness (i.e., not knowing exactly what it knows or doesn't know). Consequently, encouraging it to proceed with a step-by-step thought process seems to perplex it, leading to an inability to provide a definitive answer. Here are some sample responses: [[["question": "Sarah is a soccer fan, she never misses a match and always wears her favorite soccer jersey. Sarah is more likely to be:",         "options": {             "A": "A professional soccer player",             "B": "An accountant"         },         "answer": "B",         "deviation type": "Representativeness" "answer_GPT4_direct": "(b) An accountant",  "answer_GPT4_COT": "Sarah's love for soccer and her habit of never missing a match and always wearing her favorite soccer jersey indicates her passion for the sport. However, it does not necessarily mean she is a professional soccer player. There are many soccer fans who are not professional players. On the other hand, her being an accountant is not related to the information given. Therefore, based on the information, we cannot definitively say whether Sarah is more likely to be a professional soccer player or an accountant."  "question": "Two factories are producing light bulbs. Factory A tests a sample of 1000 light bulbs every day for defects, while Factory B tests only 100. Which factory is more likely to get exactly 10% defective light bulbs in its sample on a given day?",         "options": {             "A": "Factory A",             "B": "Factory B"         },         "answer": "B",         "deviation type": "Insensitivity to Sample Size" "answer_GPT4_direct": "(b) Factory B",         "answer_GPT4_COT": "First, we need to understand that the probability of getting exactly 10% defective light bulbs in a sample is influenced by the size of the sample. \n\nIn Factory A, 10% of 1000 light bulbs is 100 bulbs. So, Factory A needs exactly 100 defective bulbs out of 1000 to meet the 10% criteria.\n\nIn Factory B, 10% of 100 light bulbs is 10 bulbs. So, Factory B needs exactly 10 defective bulbs out of 100 to meet the 10% criteria.\n\nThe larger the sample size, the more likely it is to get a result that is close to the expected value (in this case, 10% defective). This is due to the law of large numbers, which states that as a sample size grows, its mean gets closer to the average of the whole population.\n\nTherefore, Factory A, with its larger sample size, is more likely to get exactly 10% defective light bulbs in its sample on a given day.\n\nSo, the answer is (a) Factory A."]]]. Due to the constraints of paper length, we have placed this portion of the response sample in Appendix D. |

---

> > > > ### Author Response · Authors · 2023-11-22
> > > >
> > > > | Comment | Response |
> > > > | --- | --- |
> > > > | Given that the improvements were not so great in GPT3.5, would you say that RoT prompting is an emergent ability? | We believe that the ROT approach contributes to the development of emerging capabilities in models. However, our experiments have also revealed that cognition is a high-level skill that is only beneficial for models with a high intelligence base. To conduct comparative research, we introduced the LLAMA2-13B-chat model into our study. This was done on the biased dataset constructed in this paper,on an aggregate level, LLAMA2-13B-chat demonstrates a higher degree of cognitive bias compared to GPT-4 and GPT-3.5-turbo under the RoT setting. Specifically, LLAMA2-13b-chat scored 41.8% lower on the cognitive bias test set than GPT-4.It is noteworthy that the performance of the LLAMA2-13B-chat model decreased by 32.8% and 12.1% after applying the CoT and RoT strategies, respectively. A similar trend was observed in the GPT-3.5-turbo model, where its performance dropped by 30.1% after the application of the CoT strategy. Upon analyzing the answers, we found that for foundation models that are not yet sufficiently powerful, employing the CoT strategy for complex cognitive reasoning can lead the model into a state of indecision, rendering it unable to provide definitive answers and consequently deteriorating its performance on biased task sets. The GPT-3.5-turbo model managed to follow complex thinking instructions like RoT to a certain extent, leading to a slight improvement in performance. However, smaller parameter models like LLAMA2-13B-chat experienced a decrease in their ability to follow instructions under the RoT thinking framework, resulting in weaker performance compared to the Direct strategy. Therefore, we believe that the RoT strategy is only effective for high-intelligence foundation models, as rational cognition is an advanced capability。 Please refer to Table 1 on page 6 of the paper for detailed experimental results data. However, we believe that the development pace of large models will not come to a halt. In the not-so-distant future, most models used by humans will gradually reach the level of GPT-4. In addition to considering aspects of model architecture from the computer science perspective, it is time for researchers to shift some of their focus towards the direction of "human-like intelligence." Rational thinking, as one of the essential modes of human intelligence, is highly suitable for testing and improving the performance of large models. Our work is merely a catalyst, and we welcome more AI researchers to start paying attention to disciplines such as cognitive science, psychology, anthropology, sociology, and others that study the human aspect. The theories in these fields are very likely to find applications in the field of AI. |
> > > > | Finally, I was wondering whether all the steps in RoT prompting are needed or whether some of them could be removed without a loss of performance. I am not asking the authors to run additional ablations but I think this is still interesting to think about. | We conducted a "step-by-step ablation" study on each component of the RoT method and performed experiments using the GPT-4 model on biased datasets to evaluate this ablation scheme.Observing the results (Table 2), it was evident that removing any step caused some detriment to the final outcomes. The most significant harm occurred with the removal of step 3, which led to a 9% decrease in model performance. This aligns with our hypothesis, as cognitive biases, from the perspectives of cognitive science and psychology, are primarily due to humans’ tendency to rely on ”System 1” for intuitive judgments, thereby lacking rigorous data analysis. Moreover, the entire RoT method is a holistic thinking framework, where each step contributes to and enhances the others. Therefore, the absence of any step inevitably leads to a decline in model performance。 In summary, based on the experimental results, it is evident that the third step plays a particularly crucial role in enhancing the model's cognitive reasoning capabilities. For specific experimental data, please refer to Table 2 on page 7 of the latest paper. |
> > > >
> > > > Above are all our responses, which have been updated in the latest version of our manuscript. For more detailed information, please refer to the newly submitted version of our paper.
> > > > Once again, we thank you for your time and effort.

---

> > > > > ### Comment · Reviewer_58ny · 2023-11-22
> > > > >
> > > > > Thanks to the authors for their reponse. I will not have the time to go through it today. I'll read it later this week and update here again if possible.

---

> > > > > > ### Author Response · Authors · 2023-11-23
> > > > > >
> > > > > > Okay,thank you for your time.   :)

---

### Official Review · Reviewer_QJRm · 2023-10-29

**Soundness:** 2 fair
**Presentation:** 3 good
**Contribution:** 3 good
**Rating:** 6
**Confidence:** 4

**Summary:**

The paper presents a prompt engineering technique, called RoT (Rationality of Thought), whose goal is to align with the underlying logic of human cognition. The paper claims that RoT can assist large language models (LLMs) to reduce cognitive biases. Compared with Chain-of-Thought and other reasoning-enhancement techniques, RoT does not need additional manual annotations. RoT integrates various methods to improve reasoning of LLMs (e.g. thought chains, self-reflection, and expert knowledge) into a unified theoretical framework rooted in cognitive psychology. As part of the study, the paper has created a cognitive bias dataset.

**Strengths:**

1. The paper proposes a new prompt engineering technique, RoT designed to avoid cognitive bias. RoT include steps: identification, decomposition, reflection, calculation, and evaluation, among other markers of rational thought.

2. When applied RoT to GPT-3.5-turbo and GPT-4 on the cognitive bias test set, accuracy improvements are 1.5% and 18.7%, respectively.

3. The paper collects a cognitive bias test dataset including 29 such biases.

**Weaknesses:**

1. There does not seem to be convincing evidence that Rational of Thought method can improve arithmetic reasoning and common sense reasoning tasks. Specifically improvements on GSM8K (Cobbe et al., 2021), SVAMP (Patel et al., 2021),AQUA-RAT (Ling et al., 2017) , and ARC (Clark et al., 2018) over CoT are +0.4, -0.6, +4.8, -0.3, +0.7. Other than AQUA-RAT, the performance degradations and improvements are all very small. However, the paper does not dive deep into AQUA-RAT performance improvement.

As shown in the paper "Large Language Models Cannot Self-Correct Reasoning Yet", it is not clear RoT can improve reasoning when compared with CoT.

2. Despite connecting RoT to human cognition, there is little insight on why RoT can improve cognitive biases.

**Questions:**

Now there are a large number of prompt engineering techniques that decompose problems into sub-problems. The paper should comment and hopefully compare to some of the most notable ones, such as Least to most prompting, tree of thoughts, etc.

Besides the GPT family models, the authors are encouraged to study Anthropic Claude and open source models such as LLaMA2. This would be very important to see whether the techniques can also apply to other LLMs.

The evaluation seems to focus on multiple choices. LLMs are known to have position bias. Does the result change if you change the ordering of the answer choices?

== post rebuttal ==

The authors have made a good effort addressing my comments. The paper has improved quite a bit. However, there are still some remaining concerns.

1. RoT zeroshot does not perform better than CoT zeroshot on GSM8K. Are there any insights?
2. The improvement over CoT for GPT4 is 3% absolute. It is still relatively small.
3. Given the negative results on LLaMA, it is important to consider Claude.

As a result, my rating is borderline, between marginally accept and marginally reject. However, I have marked my score to marginally accept since there is rating in between.

---

> ### Author Response · Authors · 2023-11-22
>
> Dear Reviewers,
>
> First and foremost, we would like to express our sincere gratitude for the time and effort you have dedicated to reviewing our manuscript.Your reviews have been extremely helpful in improving our research work. This process has not only strengthened our current manuscript but also enriched our understanding of the subject matter.
> In the past week, in reference to your reviews, we have supplemented the relevant experiments and clarified some of the previously unclear descriptions in our manuscript. Next, we will respond to your comments one by one.
>
> Please kindly note, due to the extensive nature of my response and the limitation of 5000 characters per message, I will be submitting my response in multiple parts. I kindly ask for your patience and suggest that you review them collectively once all parts have been sent. Thank you very much for your understanding.
> | Comment | Response |
> | --- | --- |
> | There does not seem to be convincing evidence that Rational of Thought method can improve arithmetic reasoning and common sense reasoning tasks. Specifically improvements on GSM8K (Cobbe et al., 2021), SVAMP (Patel et al., 2021),AQUA-RAT (Ling et al., 2017) , and ARC (Clark et al., 2018) over CoT are +0.4, -0.6, +4.8, -0.3, +0.7. Other than AQUA-RAT, the performance degradations and improvements are all very small. However, the paper does not dive deep into AQUA-RAT performance improvement. | We have added COT zero-shot experiments, and the results show that, compared to CoT zero-shot, RoT zero-shot enables the GPT-4 model to achieve improved results on the following datasets: SVAMP (+1.8), AQUA-RAT (+6), ARC-e (+3.9), and ARC-c (+4.1). This improvement is quite significant, and based on these results, we believe that the Rational of Thought method can enhance performance in arithmetic reasoning and common sense reasoning tasks. However, in our experimental results, we also observed a puzzling phenomenon: on the GSM8K dataset, both GPT-3.5-turbo and GPT-4 models performed better using the CoT zero-shot strategy compared to the CoT few-shot strategy. The specific performances are as follows: GPT-3.5-turbo:CoT-zeroshot，79.2    GPT-3.5-turbo:CoT-fewshot， 76.6 GPT-4:CoT-zeroshot，95.7 GPT-4:CoT-fewshot， 94.1 We speculate that this may be because, under the zero-shot setting, the GPT-4 model's own reasoning process on the GSM8K dataset is superior to the reasoning provided by the few-shot examples of the CoT method. Thus, adding few-shot examples actually led to a decrease in performance. If this is true, it would represent another intriguing research topic. The AQUA-RAT dataset primarily focuses on algebra problems, often centering on computation and algebraic reasoning, requiring solving equations, handling ratios, and percentages, among others. Therefore, the thinking framework of the RoT method, compared to the CoT method, seems more suited for tackling these types of problems, which we also observed from the content of the output answers.For details, please refer to Appendix D of our paper due to space limitations in the main text. |
> | As shown in the paper "Large Language Models Cannot Self-Correct Reasoning Yet", it is not clear RoT can improve reasoning when compared with CoT. | We have added a COT zero-shot experiment, which demonstrates that the RoT zero-shot method leads to improvements in the GPT-4 model's performance on several datasets: SVAMP (+1.8), AQUA-RAT (+6), ARC-e (+3.9), and ARC-c (+4.1). This improvement is quite significant. Based on these results, we believe the Rational of Thought method can improve arithmetic reasoning and common sense reasoning tasks. Specifically, we conducted an ablation study on the RoT method, removing each step one at a time, and used the GPT-4 model to experiment on biased datasets. Observing the results (Table 2), it was evident that removing any step caused some detriment to the final outcomes. The most significant harm occurred with the removal of step 3, which led to a 9% decrease in model performance. This aligns with our hypothesis, as cognitive biases, from the perspectives of cognitive science and psychology, are primarily due to humans’ tendency to rely on "System 1" for intuitive judgments, thereby lacking rigorous data analysis. Moreover, the entire RoT method is a holistic thinking framework, where each step contributes to and enhances the others. Therefore, the absence of any step inevitably leads to a decline in model performance. Overall, according to the experimental data, step 3 is particularly crucial for improving the model's cognitive reasoning abilities. For detailed experimental results, please refer to Table 2 on page 7 of the latest paper. |

---

> > ### Author Response · Authors · 2023-11-22
> >
> > | Comment | Response |
> > | --- | --- |
> > | Despite connecting RoT to human cognition, there is little insight on why RoT can improve cognitive biases. | The ROT approach integrates human rational thinking patterns into the model's thought process. It introduces a series of structured and reflective steps, which are instrumental in identifying and correcting biases. Specifically, several key aspects of rational thinking patterns are particularly important for mitigating cognitive biases:1. Rational thinking requires the model to evaluate more information and consider different dimensions, thereby reducing biases such as "confirmation bias." 2. It encourages systematically gathering and analyzing information. This approach ensures that all relevant information is considered and processed in a structured manner, reducing biases caused by selective or one-sided information.3. Rational thinking emphasizes logical reasoning and evidence-based analysis. This means that any decision or conclusion must be supported by sufficient facts and data, helping to avoid decisions based on intuition or preconceived notions.4. A crucial aspect of rational thinking is the reflection and evaluation of processes and outcomes. Such reflection helps identify potential biases in the thought process and provides opportunities for improvement in future decision-making. This explanation has been added to the latest version of the paper's introduction. |
> > | Now there are a large number of prompt engineering techniques that decompose problems into sub-problems. The paper should comment and hopefully compare to some of the most notable ones, such as Least to most prompting, tree of thoughts, etc. | The methodology in the "least to most" paper bears a strong resemblance in logic to the Chain of Thought (CoT) method, functioning through the breakdown of complex tasks into simpler steps followed by their sequential execution. The key prompt used is "Let's breakdown this problem." The "least to most" approach also requires a certain number of shots to enable significant performance improvement in the model. However, this method does not incorporate rational thinking processes such as "reflection," hence the limited improvement in many task performances. Considering time and cost constraints, we only experimented with the "least to most" method on the AQUA dataset, which contains only 254 questions. The resulting data is as follows:GPT-4:L2M zeroshot，85.0 / GPT-3.5-turbo:L2M zeroshot，74.4 . These results are nearly identical to those of the CoT zeroshot on the same dataset, aligning with our hypothesis. GPT-4:CoT-zeroshot，84.6 / GPT-3.5-turbo:CoT-zeroshot，75.2.However, as the "least to most" paper aligns with the direction of our research, we have included the key points of this work in the introduction section of our paper. The method proposed in the "tree of thoughts" paper necessitates designing a separate 'tree of thought' for different tasks. We believe this approach is not suitable for comparison with universal thinking frameworks like the RoT method. Nevertheless, since the "tree of thoughts" paper aligns with our research direction, we have showcased its key content in the introduction section of our paper. |
> > | Besides the GPT family models, the authors are encouraged to study Anthropic Claude and open source models such as LLaMA2. This would be very important to see whether the techniques can also apply to other LLMs. | We have added the LLAMA2-13B-chat model for comparative research. In the bias dataset constructed for this paper, on an aggregate level, LLAMA2-13B-chat demonstrates a higher degree of cognitive bias compared to GPT-4 and GPT-3.5-turbo under the RoT setting. Specifically, LLAMA2-13B-chat scored 41.8% lower on the cognitive bias test set than GPT-4. It is noteworthy that the performance of the LLAMA2-13B-chat model decreased by 32.8% and 12.1% after applying the CoT and RoT strategies, respectively. A similar trend was observed in the GPT-3.5-turbo model, where its performance dropped by 30.1% after the application of the CoT strategy. Upon analyzing the answers, we found that for foundation models that are not yet sufficiently powerful, employing the CoT strategy for complex cognitive reasoning can lead the model into a state of indecision, rendering it unable to provide definitive answers and consequently deteriorating its performance on biased task sets. The GPT-3.5-turbo model managed to follow complex thinking instructions like RoT to a certain extent, leading to a slight improvement in performance. However, smaller parameter models like LLAMA2-13B-chat experienced a decrease in their ability to follow instructions under the RoT thinking framework, resulting in weaker performance compared to the Direct strategy. Therefore, we believe that the RoT strategy is only effective for high-intelligence foundation models.Please refer to table 1 on page 6 of the paper. |

---

> > > ### Author Response · Authors · 2023-11-22
> > >
> > > | Comment | Response |
> > > | --- | --- |
> > > | The evaluation seems to focus on multiple choices. LLMs are known to have position bias. Does the result change if you change the ordering of the answer choices? | We have added an ablation study that involves shuffling the order of answers. When we changed the order of answers in the biased dataset, the output of the model varied. Specifically, under the GPT-4 zero-shot setting, the results for the three methods were as follows: GPT-4:Direct zeroshot，72.6 （-6.1，-7.8%）/ GPT-4:CoT zeroshot，60.8 （-6.2，-9.3%）/ GPT-4:RoT zeroshot，90.1（-7.3，-7.5%）/ In contrast, without shuffling the order of answers, the original results were: GPT-4:Direct zeroshot，78.7 / GPT-4:CoT zeroshot，67.0 / GPT-4:RoT zeroshot，97.4  / It can be observed from the results that shuffling the order of answers led to a general decline in model performance, but the proportion of decline was similar across methods, implying that the RoT approach still retains a significant advantage. Due to the length constraints of the paper, the results of this ablation study have been included in Appendix A. |
> > >
> > > Above are all our responses, which have been updated in the latest version of our manuscript. For more detailed information, please refer to the newly submitted version of our paper.
> > > Once again, we thank you for your time and effort.

---

### Official Review · Reviewer_Zkyz · 2023-11-01

**Soundness:** 4 excellent
**Presentation:** 3 good
**Contribution:** 3 good
**Rating:** 8
**Confidence:** 5

**Summary:**

This paper proposes Rationality of Thought, a prompting technique for reducing cognitive biases in LLMs. Based on a diverse collection of psychology papers, the authors compose a dataset of 464 questions reflecting 29 cognitive bias types. The RoT prompt improves performance of GPT-3.4 and -4 on this set of questions, and transfers effectively to other LLM benchmarks.

**Strengths:**

- The approach is well grounded in studies of human psychology.
- The authors expend significant effort on summarizing the cognitive bias types in prior psychology work.
- The constructed dataset is valuable for assessing the cognitive biases of general-purpose AI agents.
- The RoT prompt is simple and effective. It improves the performance of LLMs significantly, outperforming chain-of-thought.

**Weaknesses:**

- The data collection process is insufficiently detailed. See questions.
- The prompt search process is not transparent. See questions
- Missing ablation studies.
- The authors do not discuss the limitations of this approach (e.g., time/#tokens, generalizability).

**Questions:**

- Does the dataset have a train and a test set? If yes, are the results in figure 2 on the train or test set? If no, did you search for a prompt to overfit GPT-4 on the dataset?
- The questions are collected from "multiple authoritative psychological works"? Are they the same as those from which you compose the list 93 common cognitive biases? If not, you must cite them explicitly.
- What are the limitations of the dataset? Is it also biased in some way (e.g., representing only Western culture, focusing on a majority group)?
- How did you come up with the prompt? Are these steps of solving a problem proposed prior psychology work?
- For a fair comparison, can you also provide the results of CoT zeroshot in table 3? It is possible to put the average cost of each prompting technique as a column?
- Can you conduct an ablation study to demonstrate the importance of each step in RoT?
- Step 3 of the prompt encourages "probability calculations, Bayesian methods, and other data analysis techniques". It is a very specific request. Do you have evidence that the models actually follow this instruction? How do they behave when the solution of a problem does not require one of those techniques?
- Do you use the same prompt for each evaluation domain? Are there any customizations?
- Do you specify all the RoT steps all at once or input one step, wait for GPT to respond and then input the next step?
- It would be more interesting to try this method on tasks on which the Direct-zeroshot performance is low (e.g., code generation?). Except GSM8K, performance of GPT-4 on other tasks is already strong.

---

> ### Author Response · Authors · 2023-11-22
>
> Dear Reviewers,
>
> First and foremost, we would like to express our sincere gratitude for the time and effort you have dedicated to reviewing our manuscript.Your reviews have been extremely helpful in improving our research work. This process has not only strengthened our current manuscript but also enriched our understanding of the subject matter.
> In the past week, in reference to your reviews, we have supplemented the relevant experiments and clarified some of the previously unclear descriptions in our manuscript. Next, we will respond to your comments one by one.
>
> Please kindly note, due to the extensive nature of my response and the limitation of 5000 characters per message, I will be submitting my response in multiple parts. I kindly ask for your patience and suggest that you review them collectively once all parts have been sent. Thank you very much for your understanding.
>
> | Comment | Response |
> | --- | --- |
> | Does the dataset have a train and a test set? If yes, are the results in figure 2 on the train or test set? If no, did you search for a prompt to overfit GPT-4 on the dataset? | Our bias dataset was not divided into training and testing sets. After constructing this dataset, we directly tested the models (GPT-4, GPT-3.5-turbo, LLAMA2-13B-chat) on it, using the RoT strategy. Therefore, the results in Figure 2 can be understood as the performance of these models on an unseen dataset.  Additionally, we did not search for a prompt to overfit GPT-4 on the dataset. Our RoT strategy is grounded in professional logic from cognitive science and psychology. Specifically, our approach references works such as "Thinking, Fast and Slow" by Daniel Kahneman, "The Rationality Quotient: Toward a Test of Rational Thinking" by Keith E. Stanovich et al., "Beyond IQ: A Triarchic Theory of Human Intelligence" by Robert J. Sternberg, and "Intuition Pumps And Other Tools for Thinking" by Daniel Dennett, among others. Drawing from these comprehensive theories in cognitive science and psychology, we decompose the process of rational thinking into six steps, which are the core of our proposed RoT method:   1. Identify a complex or novel task: Recognize the essence and key issues of the current task. 2. Mobilize cognitive and physical resources: Utilize the large model for accessing existing concepts, data, and solutions relevant to the task. 3. Formulate solutions: Establish objectives and select appropriate cognitive strategies (e.g., employing prior probabilities and Bayesian methods for inferential reasoning). 4. Analyze data, execute logical reasoning: Monitor and review each step in real-time during the task execution. 5. Generate solutions or decisions: Compute the likelihood of each potential answer and then provide the most probable one. 6. Evaluate output results and make corresponding adjustments: Assess from a rational standpoint and provide the final answer. These steps synthesize key concepts and theories from the aforementioned works on human rational thinking. For example:   - Identifying complex or novel tasks: This step relates to the concept of "bounded rationality" introduced by Herbert Simon in "The Bounds of Reason." Simon emphasized the limitations and necessity of simplification strategies in human information processing, echoing the process of recognizing and understanding novel tasks. - Formulating solutions: Keith E. Stanovich's "Rational Choice: The Psychology of Rational Thought" offers an in-depth analysis of using different cognitive strategies for problem-solving, consistent with setting goals and choosing strategies in solution formulation. - Analyzing data, executing logical reasoning: The description of System 2 in Daniel Kahneman's "Thinking, Fast and Slow" fits this step well. System 2 involves slow, effortful logical thinking, closely associated with data analysis and logical reasoning execution, etc. Thus, we constructed the RoT method directly through these key concepts and theories, a method proven effective through empirical testing. |
> | The questions are collected from "multiple authoritative psychological works"? Are they the same as those from which you compose the list 93 common cognitive biases? If not, you must cite them explicitly. | Yes，They are the same as those from which we compose the list 93 common cognitive biases.We have already made a clear reference list behind"multiple authoritative psychological works".  |

---

> > ### Author Response · Authors · 2023-11-22
> >
> > | Comment | Response |
> > | --- | --- |
> > | What are the limitations of the dataset? Is it also biased in some way (e.g., representing only Western culture, focusing on a majority group)? | We believe the limitations of this bias dataset are as follows: it encompasses a relatively limited variety of types, and does not cover cognitive biases comprehensively. Additionally, the number of questions corresponding to each bias type is also limited. The seed questions of this dataset are derived from authoritative research in cognitive science, psychology, and related fields. These studies focus on the patterns of human thought, exploring the underlying logic of human intelligence, rather than merely investigating the behaviors of specific demographic groups. The data used in these studies span multiple regions and diverse populations. Therefore, the test questions in this dataset do not involve biases that "only represent Western culture" or are "focused on the majority group".  Due to space constraints in the paper, we have included the explanation of the dataset's limitations in Appendix A. |
> > | How did you come up with the prompt? Are these steps of solving a problem proposed prior psychology work? | The concept behind our study originated from applying pretrained large models to tasks in the domain of behavioral finance. Tasks in behavioral finance require models to possess a high degree of rationality, as only rational models can ensure the absence of cognitive biases in their outputs. To enhance the model's rationality, we noted that many studies in behavioral finance also draw upon theories from cognitive science and psychology. These interdisciplinary references guided us to the strategy of rational thinking prompts that we propose in our work. |
> > | For a fair comparison, can you also provide the results of CoT zeroshot in table 3? It is possible to put the average cost of each prompting technique as a column? | Yes, we have now added the results for CoT zero-shot and included these in Table 3. To facilitate comparison for readers, we have also introduced a column for averages in Table 3, showcasing the mean performance of each method. The latest results indicate that, under a zero-shot setting, the RoT method significantly outperforms the CoT method. Specifically, on several datasets, the average performance of GPT-3.5-turbo has improved by 1.5%, and GPT-4's average performance has seen a 3% increase. For detailed experimental result data, please refer to Table 3 on page 9 of the updated manuscript. |
> > | Can you conduct an ablation study to demonstrate the importance of each step in RoT? | We have conducted an ablation study on the "remove-one-at-a-time" (RoT) method, examining each step individually, and employed the GPT-4 model to test this ablation scheme on a biased dataset. Observing the results (Table 2), it was evident that removing any step caused some detriment to the final outcomes. The most significant harm occurred with the removal of step 3, which led to a 9% decrease in model performance. This aligns with our hypothesis, as cognitive biases, from the perspectives of cognitive science and psychology, are primarily due to humans’ tendency to rely on “System 1” for intuitive judgments, thereby lacking rigorous data analysis. Moreover, the entire RoT method is a holistic thinking framework, where each step contributes to and enhances the others. Therefore, the absence of any step inevitably leads to a decline in model performance. Overall, based on the experimental data, step 3 is particularly crucial for improving the model's cognitive reasoning capabilities. For specific experimental result data, please refer to Table 2 on page 7 of the latest paper. |

---

> > > ### Author Response · Authors · 2023-11-22
> > >
> > > | Comment | Response |
> > > | --- | --- |
> > > | Step 3 of the prompt encourages "probability calculations, Bayesian methods, and other data analysis techniques". It is a very specific request. Do you have evidence that the models actually follow this instruction? How do they behave when the solution of a problem does not require one of those techniques? | We have examined the model's output answers and indeed observed that the model follows the third step when answering questions, especially the GPT-4 model. Here is a specific example of an answer:   [ "question": "Which of the following investment choices would you prefer? A: Investing in a low-risk business with an annual return of 8% B: Investing in a high-risk start-up with an potential annual return of 20%.",         "options": {             "A": "Invest in low-risk business",             "B": "Invest in high-risk start-up"         },         "answer": "B",         "deviation type": "Risk aversion",  "answer_GPT4_ROT_base123456": "1. The cognitive bias involved in this task could be \"Risk Aversion\" or \"Overconfidence Bias\". Risk aversion is the behavior of humans, when exposed to uncertainty, to attempt to reduce that uncertainty. Overconfidence bias is a well-established bias in which someone's subjective confidence in their judgments is reliably greater than their objective accuracy. The impact of these biases can lead to suboptimal decision making, particularly in the field of investment. Existing correction methods involve awareness and education about these biases, and the use of statistical thinking and probability-based decision making.\n\n2. The primary approach to address this problem would be to use Bayesian methods and probability calculations to objectively assess the expected return of each investment choice. Detailed steps would include:\n   - Identify the prior probabilities of success and failure for each type of investment based on historical data or expert opinion.\n   - Calculate the expected return for each investment choice.\n   - Compare the expected returns and choose the option with the highest expected return.\n\n3. Let's assume the prior probabilities of success for the low-risk business and high-risk start-up are 0.95 and 0.6 respectively. The prior probabilities of failure would then be 0.05 and 0.4 respectively. The expected return for each investment choice can be calculated as follows:\n   - Expected return for low-risk business = 0.95 * 8% = 7.6%\n   - Expected return for high-risk start-up = 0.6 * 20% = 12%\n\n4. Reflecting on these calculations, they seem reasonable given the assumptions made. However, it's important to note that these are expected returns and actual returns could be different due to various factors.\n\n5. The plausibility of each alternative option depends on the risk tolerance of the individual. If the individual is risk averse, they might prefer the low-risk business despite the lower expected return. If the individual is risk tolerant, they might prefer the high-risk start-up due to the higher expected return.\n\n6. Based on the results of the calculations, the high-risk start-up has a higher expected return.\n\n7. #### The chosen option is (b) Invest in high-risk start-up. However, this decision should be made in the context of the individual's risk tolerance and overall investment strategy."   ]   In the above-mentioned ablation study, when the third step was removed, the model's performance significantly declined, the model will fall into a situation where it cannot make decisions. According to the experimental result data, the third step is particularly important for improving the model's cognitive reasoning abilities. Due to the paper's length constraints, this part of the answer samples has been placed in Appendix D. |

---

> > > > ### Author Response · Authors · 2023-11-22
> > > >
> > > > | Comment | Response |
> > > > | --- | --- |
> > > > | Do you use the same prompt for each evaluation domain? Are there any customizations? | Yes, we essentially used the same prompt for all datasets. Specifically, the prompt content for the bias dataset is as follows: {Follow the steps below for analysis and answer the questions: 1. Based on the content of this task, first diagnose the type of cognitive bias that may be involved in this task, and then review related research to understand the cause, impact and existing correction methods of this cognitive bias. 2. Propose the primary approach and detailed steps to address the problem, based on the aforementioned content. 3. Begin executing each step. Throughout the process, prioritize utilizing probability calculations, Bayesian methods, and other rational data analysis techniques. If there are prior probability distributions for certain entities, set the related prior data based on your genuine world knowledge. 4. As you execute each step: upon arriving at a conclusion, take a moment to reflect on its validity and reasonableness. 5. Evaluate the plausibility of each alternative option. 6. Based on the results of your calculations, provide your final answer. 7. In any case, you must select one of the given options, using the format:  \#\#\#\#The chosen option is:}  The other datasets (GSM8K, SVAMP, etc., a total of five datasets) use the same prompt content, as follows: {Follow the steps below for analysis and answer the questions: 1. Based on the content of this task, first diagnose the inherent nature of the potential issues within the task, then review related studies to understand the origins, impacts, and existing solutions of the problem. 2. Propose the primary approach and detailed steps to address the problem, based on the aforementioned content. 3. Begin executing each step. Throughout the process, prioritize utilizing probability calculations, Bayesian methods, and other rational data analysis techniques. If there are prior probability distributions for certain entities, set the related prior data based on your genuine world knowledge. 4. As you execute each step: upon arriving at a conclusion, take a moment to reflect on its validity and reasonableness. 5. Evaluate the plausibility of each alternative option. 6. Based on the results of your calculations, provide your final answer. Please present your answer in the format “The answer is”} From the content of these two prompts, it is evident that almost identical prompts are used, with no other customized settings. |
> > > > | Do you specify all the RoT steps all at once or input one step, wait for GPT to respond and then input the next step? | We input all steps of the ROT in one go to the model. |
> > > > | It would be more interesting to try this method on tasks on which the Direct-zeroshot performance is low (e.g., code generation?). Except GSM8K, performance of GPT-4 on other tasks is already strong. | We acknowledge the significance of your suggestion, and therefore, we plan to incorporate this aspect in our subsequent research. We apologize, but due to time and cost constraints, we have not included other tasks with lower zeroshot performance in our supplementary experiments this time. |
> > > >
> > > > Above are all our responses, which have been updated in the latest version of our manuscript. For more detailed information, please refer to the newly submitted version of our paper.
> > > > Once again, we thank you for your time and effort.

---

> > > > > ### Comment · Reviewer_Zkyz · 2023-11-22
> > > > > **Thanks**
> > > > >
> > > > > I have raised the score to 8.

---

> > > > > > ### Author Response · Authors · 2023-11-22
> > > > > > **Thank you~~~**
> > > > > >
> > > > > > Once again,thank you very much for your insightful comments and suggestions on our paper. Your expert guidance played a pivotal role in enhancing our work, making it more rigorous and comprehensive. We sincerely appreciate the time and effort you invested in reviewing our paper. :)

---

### Official Review · Reviewer_BFjh · 2023-11-01

**Soundness:** 3 good
**Presentation:** 3 good
**Contribution:** 3 good
**Rating:** 6
**Confidence:** 4

**Summary:**

The authors propose a new prompt for LLM called "Rationality of Thought" especially designed to extract more rational answers from input queries. The prompt is evaluated on a proposed cognitive bias dataset as well as a number of reasoning datasets and is shown to outperform both direct prompting as well as the existing "chain-of-thought" prompting.

**Strengths:**

The paper is easy to read and the overall presentation is clear.

The contribution of this work -- the RoT prompt, seems novel and works especially well for super large model like GPT-4.

**Weaknesses:**

I think the only missing element is an ablation study. Given the rather wordy prompt, one wonders which part of it contributes the most to improving answer quality.

Although it is in the appendix, I am not sure what role Table 4 plays in this work. Where is this algorithm used?

**Questions:**

See above.

---

> ### Author Response · Authors · 2023-11-22
> **An ablation study on the RoT method using GPT-4 revealed that each step is crucial for performance, with step 3 being especially vital for enhancing the model's cognitive reasoning abilities, as evidenced by a significant performance drop when removed.**
>
> Dear Reviewers,
> First and foremost, we would like to express our sincere gratitude for the time and effort you have dedicated to reviewing our manuscript.Your reviews have been extremely helpful in improving our research work. This process has not only strengthened our current manuscript but also enriched our understanding of the subject matter.
> In the past week, in reference to your reviews, we have supplemented the relevant experiments and clarified some of the previously unclear descriptions in our manuscript. Next, we will respond to your comments one by one.
> | Comment | Response |
> |---------|----------|
> | I think the only missing element is an ablation study. Given the rather wordy prompt, one wonders which part of it contributes the most to improving answer quality. | We have conducted an ablation study on the RoT method by "removing one at a time" each step and utilized the GPT-4 model to experiment on a biased dataset. Observing the results (Table 2), it was evident that removing any step caused some detriment to the final outcomes. The most significant harm occurred with the removal of step 3, which led to a 9% decrease in model performance. This aligns with our hypothesis, as cognitive biases, from the perspectives of cognitive science and psychology, are primarily due to humans’ tendency to rely on ”System 1” for intuitive judgments, thereby lacking rigorous data analysis. Moreover, the entire RoT method is a holistic thinking framework, where each step contributes to and enhances the others. Therefore, the absence of any step inevitably leads to a decline in model performance. Overall, according to the experimental data, the third step is particularly crucial in improving the model's cognitive reasoning abilities. For specific experimental results data, please refer to Table 2 on page 7 of the latest paper. |
> | Although it is in the appendix, I am not sure what role Table 4 plays in this work. Where is this algorithm used? | This code illustrates the decision-making and reasoning process of the ROT algorithm, including the prior probability estimation of options, the calculation process of Bayesian probability, and the probability calculation of each option. |
>
> Above are all our responses, which have been updated in the latest version of our manuscript. For more detailed information, please refer to the newly submitted version of our paper.
> Once again, we thank you for your time and effort.

---

> > ### Comment · Reviewer_BFjh · 2023-11-22
> >
> > Thanks for your effort in preparing the revisions, I maintain my rating.

---

> > > ### Author Response · Authors · 2023-11-22
> > >
> > > Once again, our sincere thanks.

---

### Meta-Review · Area_Chair_JKcy · 2023-12-14

**Metareview:**

The paper proposes a new approach called Rationality of Thoughts (RoT) for prompting LLMs in stages so that it reduces model cognitive bias, which is demonstrated on a new dataset that the authors collected. Subsequently, RoT was tested on a few complex reasoning tasks including GSM8K, Aqua-RAT, SVAMP and outperformed the CoT baseline on most of the tasks.

While one reviewer votes for `borderline` (`QJRm`), the other reviewers appear to support the paper's acceptance.
Reviewer `QJRm` poses the question of why the accuracy results of RoT are mixed. That is, with GPT-4, RoT underperforms CoT on GSM8K and outperforms CoT by a few percent points on SVAMP, AQUA-RAT, ARC-e, and ARC-c. with LLAMA, RoT underperforms the two baselines on the cognitive bias task proposed.

I agree with reviewer `QJRm` that the benefit of RoT has not been clearly demonstrated and well understood because of the following:
1. I agree the result is mixed and the rebuttal to this question has not fully addressed the question. That is on LLAMA, RoT performs worse than CoT and zero-shot baselines. On GPT-3.5/GPT-4, RoT underperforms the two baselines on GSM8K.
2. the paper only tested RoT in the zero-shot setting. Instead, it'd be more illuminating if RoT is tested in the few-shot settings and compared against more state-of-the-art methods (Tree of Thoughts [1] and Self-Consistency with CoT [2] and optionally Instruction Tuning).
3. the authors wrote in the paper (e.g. Section 2 and Related work section) that the steps in RoT are psychologically inspired, but provided no **concrete reference** for this 6-step approach in psychology.
4. the prompts for each step in RoT are pretty long and the space of optimizing/trying different wording/text has not been tested (Table 8).

Overall, I think the work is not yet ready for publication at ICLR. And I encourage the authors to keep up the good work to improve it!


-----

[1] Tree of Thoughts: Deliberate Problem Solving with Large Language Models. NeurIPS 2023

[2] Self-Consistency Improves Chain of Thought Reasoning in Language Models. ICLR 2023

**Justification For Why Not Higher Score:**

1. The paper contains mixed results, to which there are not explanations or insights provided.
2. The motivation of the 6 steps is not concretely grounded in psychology

**Justification For Why Not Lower Score:**

N/A

---

### Decision · Program_Chairs · 2024-01-16

Reject